# Solidification of Earth's mantle led inevitably to a basal magma ocean

Charles-Édouard Boukaré[1,2 ✉], James Badro[1,3] & Henri Samuel[1,3]

One of the main interpretations of deep-rooted geophysical structures in the mantle[1] is that they stem from the top-down solidification of the primitive basal magma ocean of Earth above the core[2–6]. However, it remains debated whether solids first formed at the bottom of the mantle, solidifying upward, or above the melts, solidifying downward. Here we show that gravitational segregation of dense, iron-rich melts from lighter, iron-poor solids drives mantle evolution, regardless of where melting curves and geotherms intersect. This process results in the accumulation of iron-oxide-rich melts above the core, forming a basal magma ocean. We numerically model mantle solidification using a new multiphase fluid dynamics approach that integrates melting phase relations and geochemical models. This enables estimating the compositional signature and spatial distribution of primordial geochemical reservoirs, which may be directly linked to the isotopic anomalies measured in Archean rocks[7–11]. We find that a substantial amount of solids is produced at the surface of the planet, not at depth, injecting geochemical signatures of shallow silicate fractionation in the deep mantle. This work could serve as a foundation for re-examining the intricate interplay between mantle dynamics, petrology and geochemistry during the first thousand million years of the evolution of rocky planets.

Isotopic anomalies in short-lived radiogenic isotopic systems in mantle rocks that record magmatic differentiation processes occurring in the first 100 Myr show that the mantle of Earth has preserved chemical heterogeneities[7–11] dating back to its infancy. These findings are corroborated by the noble gas geochemical record that argues for the preservation of these early-formed geochemical reservoirs[12–14]. The solidification of a deep primitive magma ocean alone can explain this early silicate differentiation event[8,9]. This solidification process can also explain the current seismic structure of the deep mantle, in which large low shear velocity provinces (LLSVPs) and ultralow velocity zones can be interpreted as residual products of primordial magma ocean solidification[2–5]. As remnants of magma ocean solidification, the two antipodal deep-rooted (above the core–mantle boundary) LLSVPs[1] must play a leading part in global mantle and core dynamics[15], plate tectonics and hot-spot magmatism[16–19] during the entire history of the Earth. Therefore, understanding magma ocean solidification from a dynamical and petrological point of view is essential for our comprehension of the long-term evolution of the mantle of Earth and its present-day state.

These geochemical and seismological observations indicate that the last remnants of the terrestrial magma ocean were located deep in the mantle, above the core–mantle boundary, but this remains debated both dynamically and petrologically[3,20–24]. Classical magma ocean solidification models, similar to those developed for the Moon, stipulate that in a cooling magma ocean, the first solids appear at the bottom of the mantle because of the intersection of liquidus and adiabat at depths, pushing the residual melt upwards[21,25]. As crystallization proceeds, solidification is expected to occur from the bottom (core–mantle boundary) upwards. An alternative scenario, based on the fact that the solids are more buoyant than the melt in the deep mantle, argues that solidification takes place in the middle of the magma ocean, separating it into basal and shallow magma oceans[3,20,23,26]. The shallow magma ocean solidifies upwards quickly because of efficient cooling at the surface, whereas the basal magma ocean solidifies slowly, pushing the residual meltdown to the core–mantle boundary.

The issue of top-down versus bottom-up solidification is ultimately controlled by thermodynamic properties that determine (1) where solidification takes place, that is, the intersection of liquidus with temperature, and (2) where solids and liquids accumulate, that is, the density contrast between melt and solids. Moreover, the efficiency of solid–liquid phase separation plays a fundamental part because no fractionation can occur if melts cannot gravitationally segregate from solids, regardless of the petrological or geochemical nature of solids and melts. Depending on the efficiency of solid–liquid phase separation[27], the magma ocean can freeze as a homogeneous silicate reservoir (batch crystallization scenario) or form strongly fractionated reservoirs of distinct compositions (fractional crystallization scenario)[25].

## Magma ocean dynamics and multiphase flow

Previous fluid dynamics studies mostly focused on the issue of solid–liquid phase separation in magma oceans at high melt fraction[25,28–32]. The question of whether crystals settle or are suspended by the flow has been investigated. If crystals can settle; they are deposited at the

[1]Université Paris Cité, Institut de Physique du Globe de Paris, CNRS, Paris, France. [2]Department of Physics and Astronomy, York University, Toronto, Ontario, Canada. [3]These authors contributed equally: James Badro, Henri Samuel. ✉e-mail: boukare@yorku.ca

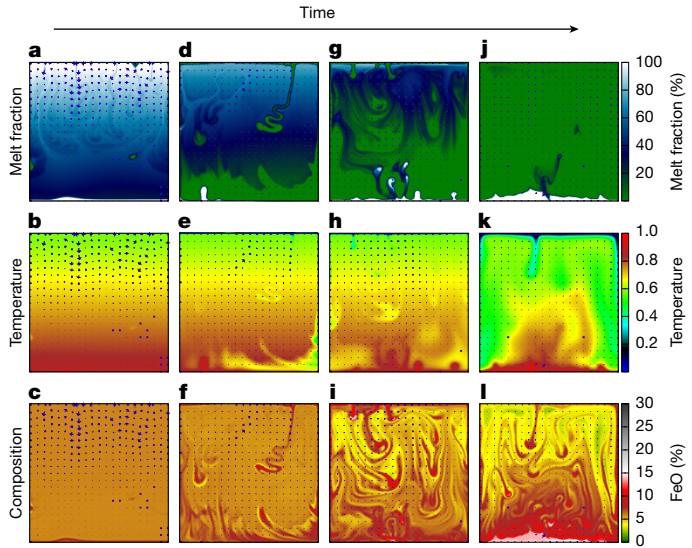

**Fig. 1 | Numerical simulations of the solidification of the mantle of Earth from a mushy magma ocean state.** Links to the associated videos can be found in the Supplementary Information. **a–c**, Initial stage: after a rapid early stage of solidification, the averaged melt fraction is approximately at the rheological critical melt fraction, that is, 50%. **d–f**, Early stage: solidification of the mush occurs in concert with thermocompositional convection at the global mantle scale. Although crystals accumulate in the deep mantle, they are formed at the surface of the planet in cold downwelling plumes. **g–i**, Late stage: progressive melt extraction from the cumulates differentiates the mantle. At the end of the upper magma ocean solidification, the mantle is heterogeneous. **j–l**, Final stage: fusible iron-rich silicates progressively pile up in the deep mantle, forming a BMO by downwards melt extraction and remelting of iron-rich solids.

bottom of the magma ocean at a rate that is a function of the settling velocity corrected by a re-entrainment flux[29,33]. Alternatively, in radial thermal evolution models, crystal fraction can be determined assuming local thermodynamic equilibrium by comparing the temperature to the solidus and liquidus at a given depth; convective mixing can be captured by a Fick's law while a prescribed phase segregation flux can capture the gravitational segregation of crystals from the magma[31].

Except for a few works[34], less attention has been given to the later stages of magma ocean differentiation, in which the mantle enters a solid-like dynamic regime because of high crystal fraction. Although previous fluid dynamics simulations have investigated the issue of phase separation efficiency, they did not account for the transport of solids and liquids of distinct composition, crystal production and remelting. These assumptions prevent the possibility of scenarios in which (1) a clear distinction can be made between the location of crystal formation and that of crystal accumulation and (2) radial magma ocean structure is controlled by the location where crystals settle, rather than the location where they form.

To address these challenges, we use a new fluid geodynamical approach with the numerical code Bambari (Methods and Supplementary Information section 1). Bambari implements a multiphase flow mathematical formalism based on the averaging method[27,35–37]. We systematically explore a range of mantle differentiation scenarios through two independent mechanisms: petrological differentiation (chemical partitioning) and mechanical differentiation (solid–liquid phase separation). Our simulations account for a density crossover between melt and solid at mid-mantle depth[22,23,38] (Supplementary Information section 1). We focus on magma ocean solidification dynamics when the mantle starts to behave as a solid.

We find that an iron-rich basal magma ocean (BMO) always forms above the core–mantle boundary (Fig. 1) and that it corresponds to the last residue of mantle solidification. This finding is irrespective

of the depth at which liquidus and adiabat intersect[20,21] (that is, where solidification is expected to occur). A BMO forms at the core–mantle boundary (CMB) through the accumulation of dense, FeO-rich solids and liquids. The FeO-rich solids, having low melting points, remelt, whereas the FeO-rich melt, being denser than the solids, migrate downwards because of negative buoyancy. Both processes contribute to the development of the BMO. We first consider the least favourable scenario for producing a BMO[21], in which the adiabat intersects the liquidus at the base of the mantle. For this scenario, unidimensional radial models predict that solidification begins at the CMB and progresses upwards.

## Magma ocean evolution

The initial conditions of our simulations represent a compositionally homogeneous mantle with a uniformly distributed melt fraction of 50%. Large solid production occurs at the surface in the earliest stage of evolution (Fig. 1a,b), because the temperature at the surface of the planet drops below the solidus, regardless of where liquidus and adiabat curves intersect at depth. The crystal-rich shallow layer becomes rapidly gravitationally unstable both thermally—because it is colder than the underlying interior—and compositionally—because it is composed of negatively buoyant crystals (Supplementary Information section 2.1). This crystal-rich unstable layer then sinks as cold downwellings. However, at this point, the solids do not accumulate in the deep mantle because they progressively remelt during their descent (Fig. 1a–c).

With further cooling of the magma ocean (Fig. 1d–f), solids formed at shallow depths gradually accumulate in the lower mantle. Our simulations show that these solids, which are still fed from the surface by cold downwellings, do not remelt anymore (Fig. 1d–f). Notably, these deep-seated solids are not the result of the crystallization of deep mantle melt (where the liquidus intersects the adiabat). These solids of shallow origin carry a chemical signature produced by low-pressure fractionation. More precisely, these solids are FeO-rich and enriched in incompatible elements. This is at odds with what we would expect solely from the values of partitioning coefficients. However, crystals forming near the cold thermal boundary tend to settle, propelling FeO-rich melt upwards by mass conservation. This FeO-rich melt approaches the cold thermal boundary, cools down by diffusion and eventually solidifies. As a result, the cold downwellings are relatively enriched in FeO as well as in incompatible trace elements. For trace elements that are very incompatible at low pressures (for example, Sm, Nd, Lu, Hf and W), their relative abundances in the residual liquid remain unchanged until the system reaches a very low degree of melting (Supplementary Fig. 3). Consequently, these shallow-origin solids, formed by the rapid quenching of melt with unfractionated trace-element ratios, are also expected to exhibit compositions that remain unfractionated relative to the bulk silicate Earth. Later, we demonstrate that the accumulation of these solids partially diminishes the chemical imprint of high-pressure fractionation processes occurring in the deep mantle.

The shallow fractionation depth of the mantle of Earth is at odds with all previous assessments based on one-dimensional solidification models, highlighting the importance of lateral variations in temperature and composition that cannot be accounted for in one-dimensional models. The reason why solids do not substantially remelt during their descent is that (1) the average temperature in the mantle (once the rheological transition is reached) is lower than the liquidus at all depths; (2) the slope of the liquidus favours crystallization at larger depths; and (3) the crystal-rich downwellings sink too fast to heat up and remelt by thermal diffusion (Supplementary Information section 2.3).

The final stage involves the accumulation of denser solids sinking as downwellings in the deep mantle (Fig. 1e–f). This forms a thermal lid above the core, leading to efficient reheating of the lowermost mantle.

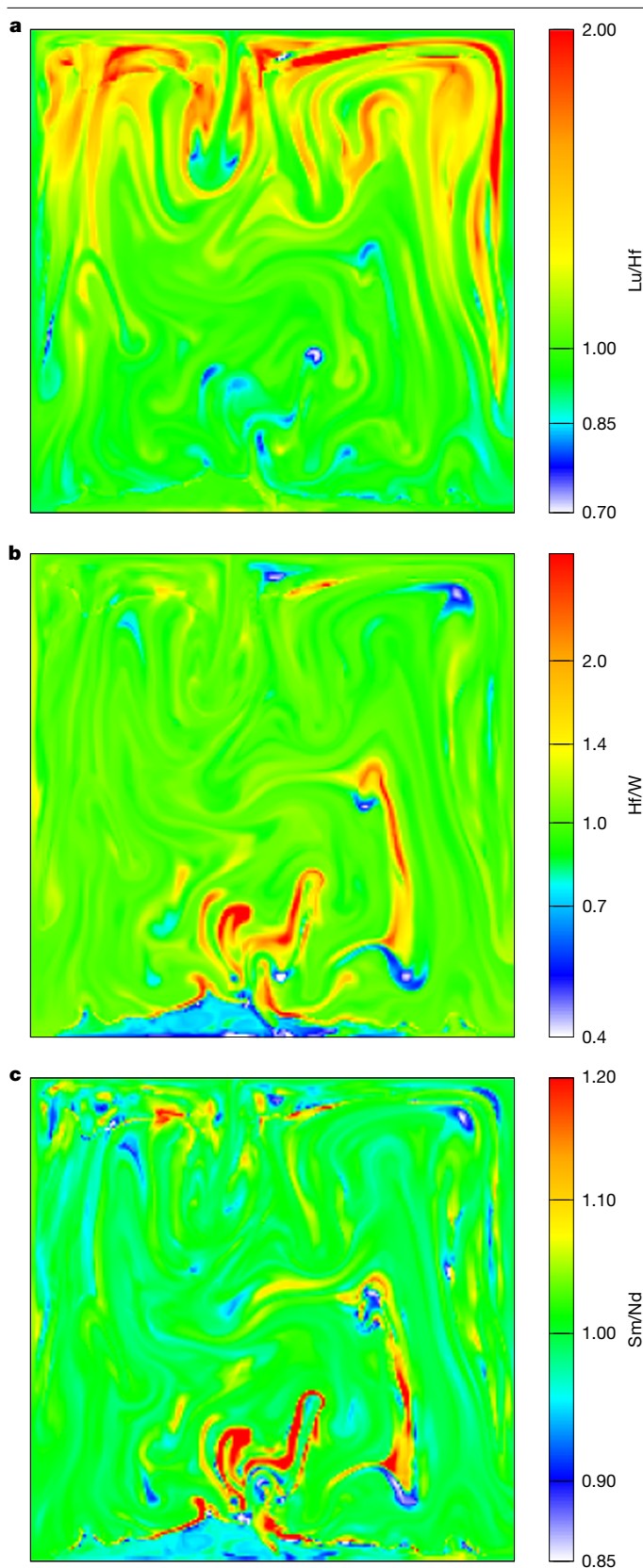

**a**

Lu/Hf

**b**

Hf/W

**c**

Sm/Nd

This in turn melts mantle fusible and FeO-rich components that are so negatively buoyant that they accumulate at the base of the mantle, forming a BMO (Fig. 1g–h). This continuous transport of solids from the surface to the lower mantle, in which they remelt and settle above the CMB, gradually enriches the BMO in FeO compared with the bulk silicate Earth (BSE).

**Fig. 2 | Geochemical signature of magma ocean solidification for an intermediate ($\delta = 10 \times 10^{-3}$) phase separation efficiency. a–c**, Snapshots at the final stage of the simulations showing the distribution of various geochemical ratios (normalized to their ratio in the BSE) across the mantle Lu/Hf (**a**), Hf/W (**b**) and Sm/Nd (**c**). Sm/Nd and Hf/W ratios are always larger than 1 in solids and lower than 1 in residual melts. By contrast, Lu/Hf behaves similarly at low pressure (olivine–melt fractionation) but becomes less than 1 in solids and greater than 1 in residual melts at high pressures (bridgmanite–melt fractionation). The geochemical signature of both low-pressure and high-pressure fractionation (Lu/Hf ratio) as well as enriched solids and depleted liquids (Sm/Nd and Hf/W ratios) are preserved in the early solid mantle after magma ocean solidification. These are then expected to be further stirred by ensuing thermochemical convection in the solid mantle throughout Earth's subsequent history. See Supplementary Figs. 6 and 7 for more extreme cases.

Let us now discuss the case of a steeper liquidus curve[20] intersecting the adiabat at mid-mantle depths, in which a BMO is the expected final stage of mantle solidification. We find that the dynamical evolution is essentially the same (Supplementary Fig. 18), which establishes that the crossing of the liquidus and adiabat does not play a main part in the style of solidification of the magma ocean. A liquidus curve that intersects the adiabat in the mid-mantle facilitates remelting in the lower mantle, promoting the formation of a BMO. Although the crystal fraction is higher in the mid-mantle, this sluggish shell does not isolate the upper mantle from the lower mantle. Downwellings that form at the surface do not accumulate at mid-mantle depth as they remain thermally negatively buoyant (Supplementary Information section 3.1 and Supplementary Fig. 18). These downwellings pursue their descent and remelt in the deep mantle.

We conclude that the formation of the basal magma ocean of Earth is primarily controlled by the relatively low melting temperature of FeO-rich silicates and the density contrast between FeO-rich liquid and coexisting solid silicates (that is, solids in thermodynamic equilibrium with that melt) and not by the relation between liquidus and adiabat. At the end of magma ocean solidification, the resulting iron-rich thermochemical structures in the lowermost mantle (Fig. 1) are geophysically and mineralogically consistent with the properties of LLSVPs (Fig. 1) and ultralow velocity zones[2–5,39,40].

## Role of solid–liquid phase separation

The solidification sequence described above requires the solid–liquid phase separation to be faster than mantle solidification and remixing by thermochemical convection. Phase separation efficiency is controlled by the dimensionless melt mobility number, $\delta$ (Supplementary Information section 2.2, Supplementary Figs. 5 and 11), which is mainly governed by crystal size and melt viscosity. In our simulations, although the viscosity of partially or essentially molten regions is unrealistically large, we preserve a realistic balance of the thermochemical convection velocity and phase separation velocity—thus, the extent of chemical differentiation—by using a relatively large melt mobility number (Supplementary Information section 2.2). To validate our approach and explore the competition between chemical differentiation and convective mixing, we explored numerically the Ra–$\delta$ parameter space, where Ra is the thermal Rayleigh number and $\delta$ is the melt mobility number. For each thermal Rayleigh number we investigated, we were able to identify a critical melt mobility number above which substantial chemical differentiation is observed (Supplementary Information section 2.2).

By extrapolating our regime diagram to magma ocean conditions, assuming a liquid silicate viscosity of 1 Pa s (refs. 41,42), melts will segregate from the solids if the crystal size is larger than 0.01 μm. Even assuming a four orders of magnitude higher viscosity ($10^4$ Pa s), the critical crystal size above which melt segregation can occur is 1 μm (Supplementary Information section 2.2), consistent with previous

estimates[25,43]. As typical grain size estimates in a magma ocean are around 1 mm (ref. 25), phase separation is expected to be efficient and dominate over remixing under realistic conditions.

A fundamental aspect is that the fluid dynamics within or in the vicinity of the top and bottom thermal boundary layers are at the heart of the chemical evolution of the magma ocean. The composition of solids formed at the surface of the planet is controlled by chemical fractionation between melt and low-pressure mineral phases. To transport a fractionated composition into the deep mantle, solids and liquids must segregate from each other in the vicinity of the cold thermal boundary (Supplementary Figs. 7 and 8). Otherwise, it is the bulk composition of the solid and liquid mixture that is brought to the deep mantle. Scaling analysis shows that the solid–liquid segregation in the top thermal boundary layer is expected to be faster than the growth rate of Rayleigh–Taylor instabilities (Supplementary Information section 2.3) in the magma ocean of Earth, allowing for chemical shallow differentiation to occur. Although present, scaling analysis indicates that this process is less pronounced in our fluid dynamics simulations than it would be under real conditions. The symmetric version of the processes described above is responsible for the formation of the BMO. FeO-rich melt migrates downwards within the hot thermal boundary layer before being entrained in upwelling currents (Supplementary Figs. 9 and 10).

## Geochemical consequences

We explore the geochemical consequences of our dynamical model on the production and nature of primordial mantle heterogeneities, inherited from magma ocean solidification. We use experimentally determined partition coefficients of trace elements between melts and the liquidus phase at upper mantle (olivine)[44] and lower mantle (bridgmanite)[45] conditions to track a simple (no garnet and no ferro-periclase) evolutionary model of key trace-elements elemental ratios (Sm/Nd, Lu/Hf and Hf/W) during solidification (Supplementary Information section 1.2). At the end of mantle solidification, solids in the upper mantle show superchondritic Lu/Hf ratios (Fig. 2a, yellow to red), owing to olivine crystallization during bottom-up crystallization of the upper mantle. Conversely, in the lower mantle above the BMO, we observe subchondritic Lu/Hf ratios (Fig. 2, blue to white) that indicate bridgmanite fractionation. Furthermore, we observe that solidification and remelting at different depths produce a complex (marble cake-like) geochemical structure in the solid mantle from core to crust (Fig. 2), far from a vision inspired by geochemical two-box modelling of single enriched (residual liquid) and depleted (precipitating solids) reservoirs[45,46]. Although the extent and magnitude of trace-element ratio distribution depend markedly on phase separation efficiency (Supplementary Figs. 6 and 7), planetary scale solidification systematically generates heterogeneities at all depths (Fig. 2).

The Lu/Hf ratio is a suitable geochemical tracer to quantify the extent of low-pressure (that is, olivine) mantle solidification on a global magma ocean[47]. This is because Lu is incompatible (that is, enriched in the melt) both in olivine and bridgmanite, whereas Hf is incompatible in olivine but compatible in bridgmanite. In this simple model, solids with superchondritic (high) Lu/Hf ratios stem from olivine crystallization and denote a shallow origin, whereas solids with subchondritic (low) Lu/Hf ratios originate from bridgmanite crystallization and can have formed only at depth. The extent of mixing between these two components on the planetary scale can be seen on a Sm/Nd-Lu/Hf correlation map (Supplementary Fig. 8). We should note, however, that the extent and amplitude of fractionation shown in Fig. 2 varies with the efficiency of solid–liquid phase separation (see Supplementary Figs. 6 and 7 for more extreme cases). The amount of stirring obtained in these two-dimensional Cartesian simulations may also be quantitatively affected by three-dimensional effects and sphericity. Unlike the lunar magma ocean, the magma ocean of Earth does not evolve as a stack of

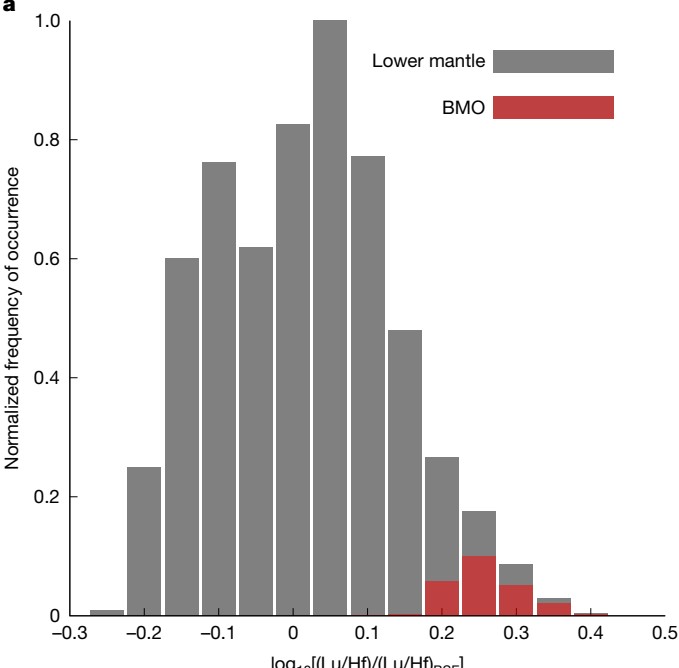

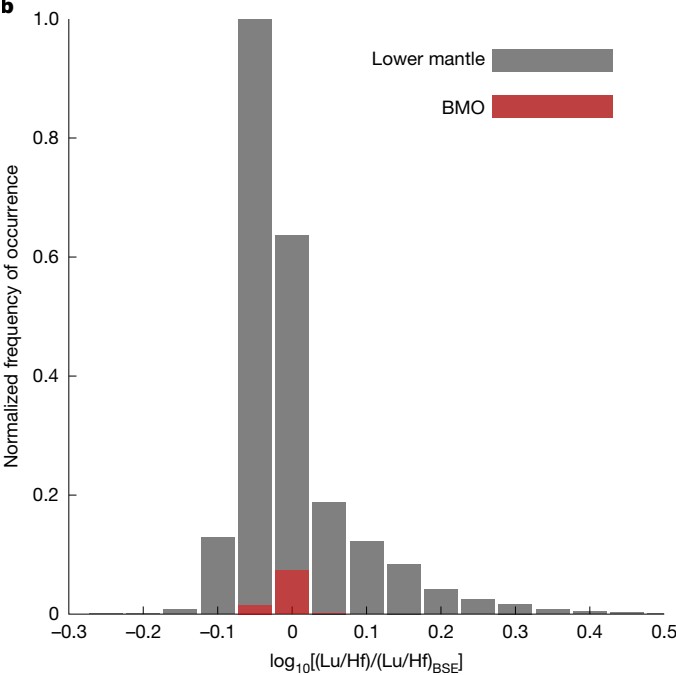

**Fig. 3 | Statistical distribution of Lu/Hf heterogeneities.** The statistical distribution is shown as the logarithm of the Lu/Hf normalized to BSE in the lower mantle (grey) and the BMO (red) at the end of magma ocean solidification. The histograms are normalized by the maximum number of occurrences. **a**, The bridgmanite–melt partition coefficients of trace elements are used at all mantle depths. **b**, The olivine–melt partition coefficients are used at low pressures and the bridgmanite–melt partition coefficients are used at high pressures. This second and more realistic case highlights the important contribution of low-pressure mineral–melt fractionation in the lower mantle and shows that bridgmanite–melt partitioning coefficients alone are not sufficient to predict the chemical evolution of the lower mantle during magma ocean solidification.

immobile layers each retaining the geochemical characteristics of the depths at which they formed. Rather, primordial heterogeneities are scattered at all depths. This marks a shift in our understanding: the way

# Article

lunar-like magma oceans evolve, based on the differentiation observed in magma chambers on Earth, does not necessarily apply to mantles as deep as the ones of the Earth or Mars. There, despite stirring during solidification caused by thermochemical convection, early-formed geochemical reservoirs are preserved but scattered throughout the mantle[48,49].

We analysed the statistical distribution of geochemical heterogeneities in the solid and molten (BMO) lower mantle, and more specifically the Lu/Hf population statistics (Fig. 3). If we account for only bridgmanite fractionation (Fig. 3a) and disregard olivine fractionation in the upper mantle, then the distribution of Lu/Hf heterogeneities is centred around 0 for the solid mantle, and the BMO composition is shifted towards higher Lu/Hf ratios (70% higher than the BSE) as we would expect from bridgmanite–melt fractionation. However, when both olivine–melt and bridgmanite–melt fractionations are accounted for (Fig. 3b), the distribution of Lu/Hf heterogeneities for the solid mantle and the BMO are both centred around 0, showing that the opposite effects of low- and high-pressure solid–melt fractionation cancel each other out.

The role of low-pressure chemical fractionation on deep mantle composition has a fundamental consequence on the quantification of the extent of magma ocean differentiation from the standpoint of trace-element ratios in mantle rocks. The primitive upper mantle (PUM) has unfractionated refractory lithophile trace-element ratios[50], and because bridgmanite crystallization strongly fractionates some of these ratios[45], this has been used to constrain the maximum amount of bridgmanite–melt differentiation that took place in the early Earth. On this basis, and to not disturb those ratios in the PUM (within their uncertainties), no more than 8% (refs. 45,51) bridgmanite can be crystallized alone in the magma ocean, which is inconsistent with experimental melt relations and the melting phase diagrams of pyrolite. Our work relaxes this constraint because bridgmanite–melt partitioning is not the only relevant process that describes the chemical differentiation between the PUM and the lower mantle during magma ocean solidification. Olivine–melt partitioning plays an important part as well, because cold downwellings form at the surface of the planet, constantly feeding the lower mantle with a low-pressure signal. These two effects have signatures that partially cancel each other out, allowing for a much larger extent of lower mantle silicate differentiation (that is, bridgmanite melt) to take place without affecting chondritic ratios in the PUM (Fig. 3).

## Magma ocean outgassing

Finally, we conducted simulations to quantify the outgassing of volatile species during magma ocean solidification (see Supplementary Information section 3.2). We used a Lagrangian approach to track and estimate the amount of mantle materials that would rise to the exsolution depth. We assumed that if mantle parcels remain below the exsolution depth, they remain undegassed and retain their primordial noble gas signature and volatile content. In agreement with previous work[52], our results indicate that a negligible fraction of the mantle can be expected to degas during magma ocean solidification beyond the rheological transition. This is consistent with the noble gas geochemical record that suggests the preservation of early-formed geochemical reservoirs[12–14], although this topic remains highly debated[53].

## Conclusions

Our modelling was performed using the least favourable conditions to produce a BMO, from both a geochemical (full equilibrium fractionation, see Supplementary Information section 1.2) and a petrological (adiabat intersects the liquidus at the base of the mantle) standpoint; yet (1) it systematically produces a BMO in the final stage of evolution, which appears to be inevitable on Earth, and extensible to other large terrestrial planets with Earth-like composition; and (2) the geochemical imprint of this solidification on the solid mantle is far less marked than that predicted by two-box geochemical models[47], owing to extensive vertical mixing during solidification. The composition of the PUM integrates the signature of a complex mixture of shallow and deep geochemical components. Concomitantly, the signature of olivine–melt trace element fractionation is present in the lower mantle at the end of magma ocean solidification and could be preserved over geological times.

These findings call for a re-interpretation of the available geochemical record and geophysical observations to better reconstruct the thermal and chemical history of Earth from its infancy to the present day, and more generally to better understand the diversity of terrestrial bodies.

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

# Article

## Methods

Recent mathematical and numerical advances[27,37,54,55] allow extending the multiphase flow formalism to geodynamic regimes relevant to magma ocean solidification dynamics with vigorous convection (expressed by the thermal Rayleigh number, which is set to $10^9$ in this work) and high melt fraction (0–100%).

We model the solid–liquid multiphase physics using the numerical code Bambari[27]. Bambari implements a multiphase flow mathematical formalism based on the averaging method[27,35–37,56–61] under infinite Prandtl number and Boussinesq approximations. The original implementation[62,63] was considerably optimized in recent years, in particular with the use of the numerical stencils implemented for the momentum conservation (equation (6)) in the Finite-Volume code StreamV[64,65]. The latter allows handling reliably large and sharp viscosity contrasts with negligible spurious pressure effects[66].

Bambari accounts for thermochemical convection in both liquid and solid states, solid–melt phase equilibrium, mineralogical phase change and solid–melt phase separation. Convective motion in a solidifying magma ocean is driven by three types of density difference. These originate from thermal expansivity (due to temperature differences), compositional differences (due to changes in iron content) and phase changes (due to the varying melt fraction)[67]. An additional mechanism that generates motion in a partially molten convective medium is the solid–liquid phase separation driven by shear deformation[68], or density contrasts between the melt and the solid. Phase separation is limited by matrix deformation, that is, compaction, and viscous drag between the melt and the solid using Darcy's law. We implemented a depth-dependent density contrast between liquids and solids[22,23,38] and used a thermodynamically self-consistent compositional evolution of melts and solids based on recent experimental melting phase diagrams[22,69] (Supplementary Information section 1 and Supplementary Figs. 1 and 2).

The model consists of two mechanical phases: a liquid phase and a solid phase. Each mechanical phase is composed of two main compositional phases corresponding to a FeO-rich end-member and a MgO-rich end-member. Trace elements are distributed among the mechanical phases according to their solid–liquid partition coefficients. Apart from the advection and diffusion of temperature, our model tracks the advection of 14 compositional fields: 7 liquids (active MgO-rich end-member, FeO-rich end-member and passive chemical species Hf, Sm, Nd, W and Lu) and the 7 solids counterparts.

In the following, we recall the governing equations solved by the code in their dimensionless form. We perform non-dimensionalization of lengths by the thickness of the mantle $H$, time by the thermal diffusion scale $H^2/\kappa$ (where $\kappa$ is the average coefficient of thermal diffusivity), velocities by $\kappa/H$, temperatures by the total super-adiabatic temperature contrast across the upper and lower thermal boundary layers in the mantle $\Delta T_\mathrm{m}$ and pressure and deviatoric stress using viscous pressure scale $\eta_s\kappa/H^2$, where $\eta_s$ is the viscosity of the solid phase.

### Mass conservation equations

The mass conservation equations for the four main chemical components are

$$\frac{\partial \phi_1}{\partial t} + \nabla \cdot \mathbf{v}_s \phi_1 = -\Gamma_1$$
$$\frac{\partial \phi_2}{\partial t} + \nabla \cdot \mathbf{v}_s \phi_2 = -\Gamma_2$$
$$\frac{\partial \phi_3}{\partial t} + \nabla \cdot \mathbf{v}_l \phi_3 = +\Gamma_1 \tag{1}$$
$$\frac{\partial \phi_4}{\partial t} + \nabla \cdot \mathbf{v}_l \phi_4 = +\Gamma_2$$

where $t$ is the time, $\mathbf{v}_l$ is the velocity of the liquid phase, $\mathbf{v}_s$ is the velocity of the solid phase and $\Gamma_i$ are associated with the rate of phase change.

The subscripts 1 and 3 refer to the MgO end-members in the solid and liquid phases, respectively. The subscripts 2 and 4 refer to the FeO end-members in the solid and liquid phases, respectively. The densities of the four chemical end-members are

$$\rho_1 = \rho_0(1 - \alpha T) + \frac{1}{2}\Delta\rho_1 - \frac{1}{2}\Delta\rho_2$$
$$\rho_2 = \rho_0(1 - \alpha T) + \frac{1}{2}\Delta\rho_1 + \frac{1}{2}\Delta\rho_2$$
$$\rho_3 = \rho_0(1 - \alpha T) - \frac{1}{2}\Delta\rho_1 - \frac{1}{2}\Delta\rho_2 \tag{2}$$
$$\rho_4 = \rho_0(1 - \alpha T) - \frac{1}{2}\Delta\rho_1 + \frac{1}{2}\Delta\rho_2,$$

where $\rho_0$ is the reference density, the thermal expansivity $\alpha$ is considered constant in the solid and liquid phases, and $T$ is the temperature. The isochemical density contrast between the liquid and solid phases is equal to $\Delta\rho_1 = \rho_1 - \rho_3 = \rho_2 - \rho_4$. Similarly, the chemical density contrast between the dense and light end-members is $\Delta\rho_2 = \rho_2 - \rho_1 = \rho_4 - \rho_3$. We parameterized these density contrasts to fit a self-consistent thermodynamic model[22] (Supplementary Fig. 1).

### Energy conservation equation

We assume local thermal equilibrium between all the phases, that is, the temperatures at a given location in all the phases are the same ($T_1(x, z, t) = T_2(x, z, t) = T_3(x, z, t) = T_4(x, z, t)$). This leads to a single equation for the conservation of energy:

$$\partial_t T + \overline{\mathbf{v}} \cdot \nabla T = \nabla^2 T - (\Gamma_1 + \Gamma_2)S_t, \tag{3}$$

where $\overline{\mathbf{v}}$ is the velocity of the solid–liquid mixture, $S_t$ is the dimensionless Stefan number, given by

$$S_t = \frac{L}{C_p \Delta T_\mathrm{m}}, \tag{4}$$

where $C_p$ is the thermal capacity at constant pressure for the mixture of the four components and $L$ is the latent heat release on phase change.

The velocity of the solid–liquid mixture is related to the solid and liquid velocities using the following equation:

$$\overline{\mathbf{v}} = \phi\mathbf{v}_l + (1 - \phi)\mathbf{v}_s. \tag{5}$$

### Momentum conservation equations

As we account for two mechanical phases, two momentum conservation equations are required to describe the two-phase flow mechanical interactions. The following Stokes equation describes the momentum conservation for the averaged solid–liquid mixture:

$$-\nabla\Pi + \nabla \cdot \underline{\underline{\tau}} - \mathrm{Ra}\, T\, \mathbf{g} - \mathrm{Rp}\, \phi\, \mathbf{g} + \frac{1}{2}\,\mathrm{Rc}\,(\phi_2 + \phi_4 - \phi_1 - \phi_3)\mathbf{g} = 0, \tag{6}$$

where $\nabla\Pi$ is the dynamic pressure gradient ($\nabla\Pi = \nabla P - (\rho_0 + 1/2\Delta\rho_1)\mathbf{g}$, where $P$ is the total pressure, $\mathbf{g}$ is the gravity vector and $\rho_0$ is the reference density) and $\underline{\underline{\tau}}$ is the deviatoric stress tensor for the solid–liquid mixture. The thermal Rayleigh number, Ra, and the phase and compositional Rayleigh numbers, Rp and Rc, respectively, are

$$\mathrm{Ra} = \frac{\rho_0 \alpha \Delta T_\mathrm{m} g H^3}{\kappa \eta_s},$$
$$\mathrm{Rp} = \frac{\Delta\rho_1 g H^3}{\kappa \eta_s}, \tag{7}$$
$$\mathrm{Rc} = \frac{\Delta\rho_2 g H^3}{\kappa \eta_s},$$

where $g$ is the constant acceleration of gravity (the magnitude of **g**). We define the stress tensor of the solid–liquid mixture as

$$\overline{\overline{\tau}} = (1 - \phi)\,\overline{\overline{\tau}}_s + \phi\,\overline{\overline{\tau}}_l = \eta(\phi)(\nabla\mathbf{v} + [\nabla\mathbf{v}]^t), \tag{8}$$

where $\eta(\phi)$ is the viscosity of fluid mixture that varies from $\eta(\phi = 0) = \eta_s$ to $\eta(\phi = 1) = \eta_l$. We use

$$\eta(\phi) = \eta_s \times 10^{\left(1 + \tanh\left(\frac{\phi - 0.5}{0.1}\right)\right)\frac{\eta_l}{2\eta_s}}. \tag{9}$$

For the second momentum conservation equation, we use a retro-action relationship[27,36] that describes $\Delta\mathbf{v} = \mathbf{v}_s - \mathbf{v}_l$, the velocity difference between the two phases:

$$\phi\Delta\mathbf{v} = \phi^2(1 - \phi)\delta^2\left(\zeta\nabla\left[\frac{(1 - \phi)}{\phi}\nabla\cdot\phi\Delta\mathbf{v}\right] + X\mathbf{g} + \nabla\cdot\overline{\overline{\tau}}\right), \tag{10}$$

where

$$X = \mathrm{Rp} + \frac{1}{2}\left(\frac{\phi_3 - \phi_4}{\phi} + \frac{\phi_2 - \phi_1}{1 - \phi}\right)\mathrm{Rc}, \tag{11}$$

where $\delta$ is the melt mobility number and $\zeta$ is the dimensionless compaction viscosity. The melt mobility number and dimensionless compaction viscosity write

$$\delta = \sqrt{\frac{r_0^2}{C_0 H^2}\frac{\eta_s}{\eta_l}},$$
$$\zeta = \frac{\eta_c}{\eta_s}, \tag{12}$$

with $r_0$ is the crystal size, $C_0$ is a constant whose value is given in the Supplementary Information and $\eta_c$ is a constant associated with the compaction viscosity, sometimes referred to as bulk viscosity, which describes the viscous resistance of a solid matrix to compact (that is, close the porous space)[35,36,70,71]. Here we follow ref. 36 in which the compaction viscosity, $\eta_{\text{bulk}}$, is defined as $\eta_{\text{bulk}} = \eta_c/\phi$ (the compaction viscosity must be infinite when the melt fraction is 0).

The above equations are discretized using conservative finite-difference schemes. The corresponding set of equations (6) and (10) involves the inversion of two sparse matrices, which are the most time-consuming operations. In Bambari, these matrix inversions are performed using the direct PARDISO library that is parallelized with OpenMP directives[72,73]. The numerical implementation of the two-phase flow physics has been benchmarked against one-dimensional analytical solutions (for example, section 3.4 in ref. 63). The new additions to the numerical implementation were successfully benchmarked against numerical and analytical solutions[65,66].

### Phase change and chemical partitioning
Methods related to the modelling of phase change and chemical partitioning can be found in Supplementary Information.

## Data availability
The raw files used to produce the figures of this paper are available at https://doi.org/10.18715/IPGP.2024.m42039nd (ref. 74). Source data are provided with this paper.

## Code availability
The code Bambari is on GitHub and available upon reasonable request.

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

**Acknowledgements** This work received funding from the European Research Council (ERC) under the Horizon 2020 research and innovation program of the European Union (grant agreement no. 101019965—SEPtiM). Parts of this work were supported by the UnivEarthS Labex program at the Université de Paris and IPGP (ANR-10-LABX-0023 and ANR-11-IDEX-0005-02), and the Natural Sciences and Engineering Research Council of Canada (RGPIN-2024-06174). We thank C. Chauvel for the discussions and K. W. Lim for his thorough proofreading of our paper. Numerical computations were performed on the S-CAPAD/DANTE platform, IPGP, France.

**Author contributions** C.-E.B. conceived and designed the research, implemented the numerical code, ran the numerical simulations, analysed the data and wrote the paper. J.B. conceived and designed the research, designed the implementation of the petrological models in the fluid dynamics simulations, analysed the data and wrote the paper. H.S. implemented and optimized the code, designed the numerical experiments, benchmarked the numerical implementation, analysed the data and wrote the paper.

**Competing interests** The authors declare no competing interests.

**Additional information**
**Correspondence and requests for materials** should be addressed to Charles-Édouard Boukaré.
