## [Peer Review File · Nature]

Solidification of Earth's mantle led inevitably to a basal magma ocean

Corresponding Author: Professor Charles-Édouard Boukaré

Version 0:

Reviewer comments:

Referee #1

(Remarks to the Author)

A. The manuscript under review describes results from a coupled model of fluid dynamics and thermochemistry of a partially solidified magma ocean that is undergoing solidification and two-phase convection. In the impressive simulations results that are presented, the authors show that a deep magma ocean can form regardless of where crystallisation occurs. It has typically been argued that a deep magma ocean would form if crystallisation of a magma ocean initiates at mid-depths rather than at the base. But the authors argue that convective overturning leads to a basal magma ocean even if crystallisation occurs upward the base. This is because in all models, lots of crystallisation occurs near the surface, where conductive cooling sharply reduces temperatures. In their models, the shallow-formed solids sink, cooling the mantle and eventually reaching the CMB. At the CMB these solids partially re-melt, creating dense melts that form a basal magma ocean. They conclude that dynamics is more important than the details of the thermochemistry (i.e., at what depth the solids crosses the adiabat).

B. This is a novel and interesting hypothesis. I am not aware of it having been discussed elsewhere. It seems plausible.

C. The study uses numerical simulations of two-phase flow, coupled with a thermochemical model of equilibrium partition coefficients as a function of pressure, temperature and composition. It also uses trace-element transport and partitioning calculations to model isotope systems. In my view, the 2D, finite volume approach used here is very well adapted to the task. The thermochemical equilibration steps, which are done by semi-coupled Picard iteration, probably don't introduce error at a significant level to affect the conclusions. Making such models work over long integration times is an impressive achievement. It is possible in part because the dimensionless numbers associated with the system have been tuned to be far away from their natural values, making the computation tractable at lower resolution in space and time.

D. The key uncertainty in these models is how they scale with the dimensionless numbers from the regime in which the calculations are made to the regime in which the natural system would be found. This is where the manuscript falls short in my view. The authors are making an argument about the dynamics: that the magma ocean can transport crystals produced near the surface to the CMB without re-melting (at least after some point in the evolution of the system). They claim that their simulations offer support for this hypothesis. But that claim is only reasonable if they can convincingly show that the results would hold, qualitatively, if they could simulate the natural conditions. They are aware of this issue and made statements along these lines. But I did not find them convincing. First of all, this is a two-phase system where segregation is essential, and they haven't really addressed this in their comments about scaling. And secondly, given the crucial importance of phase change on their results, it isn't clear whether they can make inferences from single-phase convection models.

E. In my view, higher Reynolds and Rayleigh numbers would give smaller cold plumes and more chaotic flows, which would facilitate stirring and warming of plumes, and hence remelting. So I am not confident that their results would be valid at higher Reynolds and Rayleigh numbers. Along the same lines, it isn't clear to me that the results would be the same if the simulations were initiated at lower initial melt fraction (they use a uniform melt fraction of 50% initially). And there is a potential issue in terms of segregation too: if melt viscosity is lower then crystals can segregate more easily. But this means that less heat (cold, actually) is transported by the crystals. The warmer melt they encounter as the segregate will help to melt them.

F. To improve the manuscript, I think the authors need to make a more careful analysis of the scaling of their simulations with key parameters (as noted above but also density contrast). They need to demonstrate more convincingly that their dynamical arguments hold true under the natural conditions. This can partly be done by running simulations under a range of conditions and showing the trends in results. But it would also be supported by using other methods besides just numerical solutions to extrapolate. The authors should, for example, make comparison with the results from Bower et al. <https://doi.org/10.1016/j.pepi.2017.11.004>. And it may be the case that Ra-number based flux parameterisation can also help, but these should be used with caution because they are not intended for two-phase dynamics.

G. Aside from not citing Bower et al., I think there is an avoidance of reference to the McKenzian theory for melt segregation, which is odd given that it preceded the Bercovici & Ricard papers. Contributions such as Rudge's 2011 paper (<https://doi.org/10.1111/j.1365-246X.2010.04870.x>) or Katz' 2008 paper (<https://doi.org/10.1093/petrology/egn058>) or even Ribe's 1987 paper.

H. The context provided in the manuscript is good and the lucidity is adequate. I think that it could be more lucid in explaining the key arguments in physical terms that are divorced from the numerical simulations and instead rely on scaling arguments derived (in part) from the simulations.

Referee #2

(Remarks to the Author)

Boukaré et al. model the crystallization of a magma ocean using a 2D numerical method that considers the dynamical influences of solid-liquid phase changes, FeO-induced density variations, surface cooling/convection, and tracking several key geochemical tracers. The model has been calibrated to experiments to yield plausible phase diagrams and elemental partitioning. They find that, regardless of whether crystallization of a magma ocean begins at the bottom or middle (depending on where the liquid isentrope and liquidus intersect), FeO rich material inevitably accumulates in a molten form at the base of the mantle.

I recall a speculative discussion by Labrosse et al. (2015), in a book volume edited by the second author of the present manuscript, which described a thought experiment in which the bottom-up crystallization scenario leads to this kind of outcome. It stated that "this scenario clearly needs to be explored more quantitatively once a good parameterization of the phase diagram of the mantle is known and the numerical tools required to treat it in a self-consistent way have been developed." The authors of the present manuscript have done exactly that, and more.

The "more" includes a very interesting finding (in my opinion) for how compensatory effects of shallow and deep partitioning combined with convective re-mixing can significantly mute the geochemical signature produced by large scale magma fractionation. This is the kind of finding that can only be found by conducting this level of modeling, and in my view demonstrates the intrinsic value of this kind of study. This result could be emphasized more in a minor revision.

There is one addition that might increase the value/impact of this study, which would be to follow up the subsequent sentence to the one quoted above in Labrosse et al (2015): "One drawback of this scenario is that the magma ocean at the surface is likely to strongly degas, unless thermal convection is not efficient enough to keep it well mixed or a thick and stable crust is formed at the surface of the magma ocean, and the BMO hence formed may not really explain the noble gas geochemistry." I wonder if the authors might follow this up with some comments in a minor revision, which bear upon the predicted consequences for middle-first vs bottom-first crystallization scenarios.

Version 1:

Reviewer comments:

Referee #1

(Remarks to the Author)

A. This is a resubmitted manuscript describing a novel hypothesis regarding the likelihood of a terrestrial BMO and its potential to explain mantle structure and chemistry. The hypothesis is that what matters in the formation of a BMO is not where the mantle adiabat first intersects the mantle liquidus (and hence where internal crystallisation initiates), but rather where dense, iron-rich crystals, formed near the surface by conductive cooling, ultimately settle. It further argues that the settling depth for these crystals is the CMB, where they are remelted into a dense, Fe-rich BMO. Numerical simulations tentatively support this hypothesis. The manuscript reports models of trace element ratios that are used as a proxy for the depth at which crystals form, showing that the BMO can retain the signature of shallow crystallisation and fractionation.

B. The hypothesis of this manuscript is a novel (as far as I know) and appealing idea. If correct, it is an important discovery that will be broadly impactful in Earth and planetary science.

C. The main method used in this work is numerical simulations of convection, solidification/melting, and geochemical transport. In my view, the 2D, finite volume approach used here is very well adapted to the task. The thermochemical equilibration steps, which are done by semi-coupled Picard iteration, probably don't introduce error at a significant level to affect the conclusions. Making such models work over long integration times is an impressive achievement. The simulations may be a reliable indicator of the dynamics in the regime in which they are run. However, the regime in which simulations

are run is not the regime in which the system operated -- in particular the ratio of the compaction length to the mantle height, δ , a proxy for the ease of melt segregation, is too small.

D. Again, the key uncertainty is how the models would behave under more realistic conditions. The authors have made a significant effort to address this with their new section in the appendices (1.6) on crystal remelting. I am sorry to say that I found this section unconvincing. The theory they use here is overly complicated and somewhat incorrect. The diffusive heating of a sphere is not described by an error-function similarity solution, but rather by an exponential decay (using separation of variables). The mapping of temperature change onto change in porosity can't be correct, because it is the thermal gradient that drives diffusion; converting sensible heat into latent heat modifies this and hence changes the diffusion (see Katz & Rudge G3 2011). The Stokes velocity assumes that temperature differences drive density differences, but there might be a broader argument (line 159) that the sinking crystals are chemically dense. I don't understand how the thermal Rayleigh number of the whole mantle determines the size r_0 of the blob -- it would seem that this is a consequence of delamination of a variable-viscosity "plate" and hence of the Rayleigh number of the thermal boundary layer. In the main body the sinking objects are referred to as crystals, but here they seem to be blobs. Suggestions for a simpler approach are below (F).

E. The robustness of the conclusions is a key question in evaluating this manuscript. On the one hand, the physical argument made in the hypothesis seems eminently plausible and the numerical simulations seem to support it. On the other hand, some of the key details of the numerical results are not shown or discussed, and the scaling analysis used to extend the parameter regime does not give confidence. A key issue, for me, is the formation process of the solid near the cold upper boundary. Is segregation important here, or is the solidification rapid and hence segregation unimportant? In other words, is the composition of this solid equal to the bulk composition or is it enriched in Fe? If this solid is not shaped by melt segregation, then I suppose that melt segregation near the CMB is what creates the Fe-rich BMO. So it would be good to illustrate and describe these processes in more detail.

F. Although I think the hypothesis is important, the manuscript could be much clearer about the detailed scientific logic, and its connection to what occurs in the simulations --- particularly the role(s) of melt segregation. Similarly, I would like more clarity on where `_thermal_` buoyancy is the key control, and where `_chemical_` buoyancy drives things. I think that it is (negative) thermal buoyancy that drives solid from the upper surface to the CMB. And I think that melting of these surface-produced solids at the CMB produces an Fe-enriched melt that displaces the residual solid upward. But I am not confident that this is what the authors are actually invoking, partly because these processes (particularly the melt segregation) are not illustrated in the manuscript (and partly because the text is not sufficiently clear about this). Figure 1 shows such a large-scale view of the model that what is happening in the two critical regions (sub-surface and just above the CMB) are barely visible (and there is no indication of the liquid flow field). Moreover, since these seem to be the two crucial, hypothesised processes that BMO formation relies on, they should also be investigated by scaling analysis.

For the sinking process, I now understand that the transport of solid from the upper, cold TBL to the CMB is driven by thermal (not chemical) buoyancy of a blob that is much, much larger than individual crystals (this was not clear to me on my previous review). A key parameter may therefore be the Peclet number, a ratio of a diffusion timescale to an advection timescale. The diffusion timescale is $\sim r_0^2/\kappa$, where κ might be adjusted to take account of the latent heat. The advection timescale is H/U , where U is a Stokes settling speed. The blob size r_0 might be approximated as the thickness of the delaminating part of the upper TBL (and its initial temperature and thermal buoyancy from the average temperature of this part). See for example book by Jaupart & Marcheschal. Large Pe corresponds to blobs reaching the CMB.

Another scaling analysis (or at least detailed consideration) that could/should be included addresses the segregation of melt above the CMB to form a BMO. This would be applied to the solid material that arrives at the CMB, with the question being: does melt have time to segregate (downward) before the bulk material becomes hot enough to rise?

An ambiguous point that should be clearly addressed is the density contrast between liquid and solid. In the shallow mantle, melts are less dense than solids and segregate upward. In the deep mantle and at the CMB, melts are more dense than solids and segregate downwards. There is a density crossover at about 4 GPa, I think. This aspect of the physics would seem important to the clarity of the manuscript but isn't addressed in the main text.

Here are some other places in the manuscript where I think that the writing could be improved to be clearer about the physics (by line number): 74--78, this sentence seems circular; a magma ocean solidifies from the bottom up because it solidifies from the bottom up. 81--82, this sentence is confusing because it says that efficient surface cooling and top-down solidification leads to bottom-up solidification. 151--153, by referring to crystals, this sentence seems to indicate that crystals are settling individually through a melt phase, whereas in fact it is blobs of dense solid that are sinking. 159, this reference to Fe-rich solids confused me because the baseline is unclear (Fe-rich compared to what?). I think that these solids are Fe-rich relative to the residue of melting above the CMB, but it took some thinking to realise/wonder this. 187--197, this paragraph might be referring to segregation of the Fe-rich BMO melt from the residual solids, but it isn't very specific. Also, it isn't clear whether "critical" refers to a percolation threshold or the value of a dimensionless number that controls the ratio of two processes. 242--275, this paragraph discusses both models and observations, I think, but I found it difficult to be certain I understood which was being referred to, particularly in the latter half of the paragraph. Perhaps break into two paragraphs and signpost more clearly where you refer to models versus observations.

G. References now seem appropriate.

H. See my comments in section F above. I think the lucidity of the manuscript can/should be improved if this is to be

published in Nature.

Referee #2

(Remarks to the Author)

The authors have addressed my previous comments and suggestions. There are of course still many details to be worked out in the long-term, however, I think this paper settles an important open question regarding the formation of basal magma oceans. In particular, freezing from the middle is not a requirement for the formation of such features. This dovetails nicely with other work by some of these authors on an analogous dense layer inside Mars. It may very well be that iron oxides being (1) simultaneously the heaviest major mantle component as well as (2) slightly incompatible with respect to melting causes basal magma/mush oceans to be a ubiquitous feature of all sufficiently oxidized terrestrial bodies that undergo extensive internal melting/freezing.

Version 2:

Reviewer comments:

Referee #1

(Remarks to the Author)

This is my third review of the manuscript and so I will go directly to my comments on the current version.

The manuscript is improved after the latest round of revision. I think that the idea is novel and important, in that it builds a robust physical explanation for the existence of a basal magma ocean from the dynamics of mantle convection, magma segregation, chemical fractionation, and phase change, without relying on the details of the liquidus and geotherm curves. The writing could be improved to bring the ideas into sharper focus. I think the authors could do more to provide a simple explanation for where and why chemical fractionation (i.e., Fe enrichment) occurs and how that material ends up in a basal magma ocean. For example, the abstract mentions the importance of melt segregation but doesn't note the importance of this process in the *shallow* mantle, which is the most innovative part of this paper. I attach an annotated manuscript version with some suggestions.

The scaling analyses provided in the supplementary material have evolved to be more convincing in terms of their relevance, but they still seem to provide only marginal support for the conclusions. Nonetheless, I think this is adequate given the uncertainties of the problem and general plausibility of the explanation. It might be worth spending time to tighten up the writing in the supplement too.

The authors might consider the relevance of the following to their supplementary information on model development:

<https://doi.org/10.1093/gji/ggac309>, <https://doi.org/10.1146/annurev-earth-032320-083704>,

<https://press.princeton.edu/books/hardcover/9780691176567/the-dynamics-of-partially-molten-rock>

Reviewer 1

1) *The key uncertainty in these models is how they scale with the dimensionless numbers from the regime in which the calculations are made to the regime in which the natural system would be found. This is where the manuscript falls short in my view. The authors are making an argument about the dynamics: that the magma ocean can transport crystals produced near the surface to the CMB without re-melting (at least after some point in the evolution of the system). They claim that their simulations offer support for this hypothesis. But that claim is only reasonable if they can convincingly show that the results would hold, qualitatively, if they could simulate the natural conditions. They are aware of this issue and made statements along these lines. But I did not find them convincing.*

Reply: This is a well taken point. A quantitative estimate the amount of crystal remelting during their descent, along with a discussion on how our results would scale with the relevant dimensionless numbers is actually a great idea. We therefore added an analytical estimation of the evolution of crystal fraction as function of depth for a sinking parcel (see the whole new supplementary material Section 1.6). This calculation shows that smaller amounts of remelting occurs in less viscous conditions (as expected for magma ocean conditions) despite the fact that thermal plumes are expected to be smaller. This is because, to first order, heat advection becomes considerably faster than heat diffusion at higher Rayleigh numbers. For high Rayleigh and Reynolds numbers, thermal anomalies are not expected to be destroyed or to be more easily mixed in the bulk of the convective domain. They will therefore survive at much smaller length-scales than they would do at low Rayleigh and Reynolds numbers. Note that our calculation does not account for the effect of pressure on solidification. Once the the mantle temperature reaches the liquidus just below the top thermal boundary layer, pressure would in fact favor crystallization because the slope of the adiabat is steeper than that of the liquidus (as investigated in our work).

First of all, this is a two-phase system where segregation is essential, and they haven't really addressed this in their comments about scaling. And secondly, given the crucial importance of phase change on their results, it isn't clear whether they can make inferences from single-phase convection models.

Reply: This is a fair point that made us realize that our discussion, the analytical estimates, and the way we describe phase segregation in a dimensionless fashion were not clear enough. In particular, regarding our use of scaling laws of single phase convection models: we use these relationships *only as a guide* to validate our simulations that incorporate additional processes specific to multiphase flow. Our idea is to show that the issue of solid-liquid segregation and differentiation essentially results from the competition between mean flow and phase separation velocities (see the whole new supplementary material Section 1.5, Eqs. (24) and (25)). We use a scaling law extracted from a single phase model to predict the mean flow velocity as function of the thermal Rayleigh number. We then compare this velocity to an analytical phase separation velocity extracted from our multi-phase flow model see Eq. (25) and Boukaré and Ricard (2017). **We conducted additional simulations** to better cover the range of dimensionless parameters currently accessible. The revised manuscript now builds on 22 simulations (compared to 4 in the original version of the manuscript). We compared the results of our simulations to the analytical prediction that we presented above (see new Fig. S5). Despite their simplicity, our

analytical predictions are in good agreement with our simulations, which a posteriori validates our approach.

2) *In my view, higher Reynolds and Rayleigh numbers would give smaller cold plumes and more chaotic flows, which would facilitate stirring and warming of plumes, and hence remelting. So I am not confident that their results would be valid at higher Reynolds and Rayleigh numbers.*

Reply: We have addressed this concern in our reply to the point 1 above.

Along the same lines, it isn't clear to me that the results would be the same if the simulations were initiated at lower initial melt fraction (they use a uniform melt fraction of 50% initially). And there is a potential issue in terms of segregation too: if melt viscosity is lower then crystals can segregate more easily. But this means that less heat (cold, actually) is transported by the crystals. The warmer melt they encounter as the segregate will help to melt them

Reply: Regarding the first part of this question: since our focus is mantle differentiation in a magma ocean, we chose to consider a conservative scenario where the Earth's mantle is still homogeneous when it reaches the rheological transition. We agree that a different mechanism could lead to mantle differentiation at low crystal fraction, e.g., Solomatov and Stevenson (1993c,b,a); Lavorel and Le Bars (2009); Bower et al. (2018); Monteux et al. (2023), but this remains highly debated. Therefore we focussed our research on a simple scenario where differentiation from a homogeneous magma ocean starts when the mantle reaches the rheological transition. Regarding the issue of remelting, in the early stage of magma ocean, crystals remelt if the isentrope is above the liquidus. However, the mechanism that we propose here still applies at later stages of magma ocean when the isentrope is below the liquidus at all depths (see also our reply to point 1). This is fully addressed by the new analytical model we propose, and was mentioned in the main text of our revised manuscript.

3) *To improve the manuscript, I think the authors need to make a more careful analysis of the scaling of their simulations with key parameters (as noted above but also density contrast). They need to demonstrate more convincingly that their dynamical arguments hold true under the natural conditions. This can partly be done by running simulations under a range of conditions and showing the trends in results. But it would also be supported by using other methods besides just numerical solutions to extrapolate. The authors should, for example, make comparison with the results from Bower et al. <https://doi.org/10.1016/j.pepi.2017.11.004> . And it may be the case that Ra-number based flux parameterisation can also help, but these should be used with caution because they are not intended for two-phase dynamics.*

Reply: We followed the referee's recommendations by (1) conducting additional simulations over a broader range of conditions showing that the trend remains broadly consistent with our analytical estimates (see our reply to point 2 and the newly added supplementary material Section 1.5), (2) we now propose an analytical estimation on how crystal remelting in downwelling plumes would scale with the thermal Rayleigh number (see our reply to point 1 and newly written supplementary material Section 1.6), and (3) We included a paragraph in the main text (lines 99-116 in the main text and in SI Section 1.5) as well as in the supplements to better relate our findings to the work of Bower et al. (2018) and Monteux et al. (2020).

4) *Aside from not citing Bower et al., I think there is an avoidance of reference to the McKenzian theory for melt segregation, which is odd given that it preceded the Bercovici & Ricard papers. Contributions such as Rudge's 2011 paper (<https://doi.org/10.1111/j.1365-246X.2010.04870.x>) or Katz' 2008 paper (<https://doi.org/10.1093/petrology/egn058>) or even Ribe's 1987 paper.*

Reply: We are sorry for this omission, there is no avoidance whatsoever, this was unintentional. McKenzian theory is certainly central to the development of multi-phase flow theory in geosciences. We added the aforementioned references and a few others (lines 118-120 in the main text), and thank the reviewer for pointing this out.

5) *The context provided in the manuscript is good and the lucidity is adequate. I think that it could be more lucid in explaining the key arguments in physical terms that are divorced from the numerical simulations and instead rely on scaling arguments derived (in part) from the simulations*

Reply: We have done our best to address this in our responses to points 1 and 2 above, and by all the mods and additions to the manuscript and SI resulting from this.

Reviewer 2

There is one addition that might increase the value/impact of this study, which would be to follow up the subsequent sentence to the one quoted above in Labrosse et al (2015): "One drawback of this scenario is that the magma ocean at the surface is likely to strongly degas, unless thermal convection is not efficient enough to keep it well mixed or a thick and stable crust is formed at the surface of the magma ocean, and the BMO hence formed may not really explain the noble gas geochemistry." I wonder if the authors might follow this up with some comments in a minor revision, which bear upon the predicted consequences for middle-first vs bottom-first crystallization scenarios

Reply: We thank the reviewer for suggesting this. It was something we had discussed amongst us while drafting the paper, because the question of mantle outgassing and the preservation of an undegassed/primordial noble gas reservoir is crucial, and may be strongly affected by magma ocean solidification dynamics; and furthermore it can be used to discriminate between parameters or models that do or do not preserve noble gas signatures observed in the mantle. One of the co-authors of this paper has recently investigated the conditions of degassing, and we thought we had better leave this topic aside, for two reasons:

(1) a rigorous implementation is beyond the scope of this paper (see below).

(2) the preservation of an undegassed primordial mantle reservoir has recently been questioned by noble gas geochemists themselves Parai et al. (PNAS, 2022), Péron et al. (EPSL, 2022).

Yet, since this was raised by the reviewer, and rightfully so, we abide by their suggestion and added a section to the manuscript and the SI, and were careful to list the caveats.

We performed additional simulations to assess the preservation of an undegassed mantle reservoir. We followed a similar approach as in Samuel and Salvador (2023) to model volatile exsolution in our simulations. In practice, we assigned the undegassed property to passive Lagrangian tracers initially uniformly seeded in the entire domain, and we track the minimum depth that is reached by tracers by advecting them using the divergence-free mean velocity field. These tracers are therefore representative of magma ocean parcels. If a tracer does not reach the exsolution depth, its associated material remains undegassed, *i.e.*, it retains its primordial noble gas signature. On the contrary, tracers going above the exsolution depth lose their volatiles. Our results (see newly added supplementary material Section 1.7) corroborate those of Samuel and Salvador (2023) who showed that even in a fully molten, and vigorously convecting magma ocean a large fraction of the mantle does not reach the exsolution depth, and therefore a non-negligible fraction of its volatiles can be preserved.

The caveats: It should be noted that in our case, the coexistence of the liquid and solid phases along with the possibility of multiple melting and solidification events limits the applicability of the tracer approach proposed in Samuel and Salvador (2023). For this reason, our results should be taken as a simplified model that showcases the inefficiency of mantle convective dynamics to exsolve its volatiles species. A more thorough investigation is required to finely assess the outgassing capability of mushy magma oceans to exsolve their volatiles, but this requires a complete re-implementation of the numerical model to properly track dissolved volatiles in space and time and across melting and solidification processes, which goes beyond the main purpose of our current study.

Also, as mentioned above, the preservation of a primordial and undegassed mantle reservoir has recently been questioned in Parai et al. (PNAS, 2022), which concluded that while the noble gas (He, Ne, Kr) isotopic signals collected at hotspot locations may originate from separate reservoirs. Péron et al. (EPSL, 2022) extended that logic to Xe isotopes and proposed that the mantle can actually be even more depleted than the atmosphere. Primordial mantle sources do not need to have high concentrations of volatiles, which relaxes the requirement that parts of the mantle should have never degassed at all.

Given these considerations, augmented by our simple simulations, we believe that our results are not incompatible with constraints arising from noble gas geochemistry. This has been mentioned in our revised manuscript (line 271-281).

References

- Boukaré, C.-E. and Ricard, Y. (2017). Modeling phase separation and phase change for magma ocean solidification dynamics. *Geochemistry, Geophysics, Geosystems*, 18(9):3385–3404.
- Bower, D. J., Sanan, P., and Wolf, A. S. (2018). Numerical solution of a non-linear conservation law applicable to the interior dynamics of partially molten planets. *Physics of the earth and planetary interiors*, 274:49–62.
- Lavarel, G. and Le Bars, M. (2009). Sedimentation of particles in a vigorously convecting fluid. *Physical Review E*, 80(4):046324.
- Monteux, J., Qaddah, B., and Andrault, D. (2023). Conditions for segregation of a crystal-rich layer within a convective magma ocean. *Journal of Geophysical Research: Planets*, 128(5):e2023JE007805.
- Solomatov, V. S. and Stevenson, D. J. (1993a). Kinetics of crystal growth in a terrestrial magma ocean. *J. Geophys. Res: Planets*, 98(E3):5407–5418.
- Solomatov, V. S. and Stevenson, D. J. (1993b). Nonfractional crystallization of a terrestrial magma ocean. *J. Geophys. Res: Planets*, 98(E3):5391–5406.
- Solomatov, V. S. and Stevenson, D. J. (1993c). Suspension in convective layers and style of differentiation of a terrestrial magma ocean. *J. Geophys. Res: Planets*, 98(E3):5375–5390.

Reviewer 1

A) *This is a resubmitted manuscript describing a novel hypothesis regarding the likelihood of a terrestrial BMO and its potential to explain mantle structure and chemistry. The hypothesis is that what matters in the formation of a BMO is not where the mantle adiabat first intersects the mantle liquidus (and hence where internal crystallisation initiates), but rather where dense, iron-rich crystals, formed near the surface by conductive cooling, ultimately settle. It further argues that the settling depth for these crystals is the CMB, where they are remelted into a dense, Fe-rich BMO. Numerical simulations tentatively support this hypothesis. The manuscript reports models of trace element ratios that are used as a proxy for the depth at which crystals form, showing that the BMO can retain the signature of shallow crystallisation and fractionation.*

B) *The hypothesis of this manuscript is a novel (as far as I know) and appealing idea. If correct, it is an important discovery that will be broadly impactful in Earth and planetary science.*

C. *The main method used in this work is numerical simulations of convection, solidification/melting, and geochemical transport. In my view, the 2D, finite volume approach used here is very well adapted to the task. The thermochemical equilibration steps, which are done by semi-coupled Picard iteration, probably don't introduce error at a significant level to affect the conclusions. Making such models work over long integration times is an impressive achievement. The simulations may be a reliable indicator of the dynamics in the regime in which they are run. However, the regime in which simulations are run is not the regime in which the system operated – in particular the ratio of the compaction length to the mantle height, δ , a proxy for the ease of melt segregation, is too small.*

D) *Again, the key uncertainty is how the models would behave under more realistic conditions. The authors have made a significant effort to address this with their new section in the appendices (1.6) on crystal remelting. I am sorry to say that I found this section unconvincing. The theory they use here is overly complicated and somewhat incorrect. The diffusive heating of a sphere is not described by an error-function similarity solution, but rather by an exponential decay (using separation of variables). The mapping of temperature change onto change in porosity can't be correct, because it is the thermal gradient that drives diffusion; converting sensible heat into latent heat modifies this and hence changes the diffusion (see Katz & Rudge G3 2011). The Stokes velocity assumes that temperature differences drive density differences, but there might be a broader argument (line 159) that the sinking crystals are chemically dense. I don't understand how the thermal Rayleigh number of the whole mantle determines the size r_0 of the blob – it would seem that this is a consequence of delamination of a variable-viscosity "plate" and hence of the Rayleigh number of the thermal boundary layer. In the main body the sinking objects are referred to as crystals, but here they seem to be blobs. Suggestions for a simpler approach are below (F).*

Reply: We agree that the manuscript could have been stronger in terms of extrapolating the fluid dynamics simulations to realistic conditions. To address this, we have added an entirely new section in the supplementary materials titled *Scaling Analysis and Regime Diagrams*. This section consists of three subsections: 2.1) *Chemical Differentiation in Thermal Boundary Layers*, 2.2) *Solid-Liquid Phase Separation vs. Convective Mixing*, and 2.3) *Crystal Remelting During Descent*. These subsections provide a more detailed description of the fluid dynamics processes that are central to our hypothesis. More importantly, we demonstrate in these sections why our hypothesis applies to realistic conditions.

In subsection 2.1, we perform a scaling analysis to explain why we expect solid-liquid segregation near the thermal boundary layer in both our simulations and Earth's magma ocean. In subsection 2.2, we present regime diagrams (already included in the previous version of the manuscript) covering fractional crystallization and batch solidification regimes. In subsection 2.3, we completely revised our analytical estimates of crystal remelting during the descent of cold blobs, following the model of Katz & Rudge (G3, 2011). To estimate the thickness of the upper thermal boundary layer—which serves as a good approximation for the size of these “blobs”—we now use a scaling law that accounts for strong viscosity contrasts between the surface and the interior, as described in Jaupart & Mareschal (Chap 9, 2010). Once again, we thank the reviewer for their insightful comment. In hindsight, we agree that this Section is absolutely essential for the manuscript.

We also realized that we did not use the term “crystal sinking” carefully enough. In many instances, we were in fact referring to the motion of crystal-rich regions sinking as Rayleigh-Taylor instabilities. We have revised the wording, rephrasing it as “sinking as downwellings” where appropriate.

E. The robustness of the conclusions is a key question in evaluating this manuscript. On the one hand, the physical argument made in the hypothesis seems eminently plausible and the numerical simulations seem to support it. On the other hand, some of the key details of the numerical results are not shown or discussed, and the scaling analysis used to extend the parameter regime does not give confidence. A key issue, for me, is the formation process of the solid near the cold upper boundary. Is segregation important here, or is the solidification rapid and hence segregation unimportant? In other words, is the composition of this solid equal to the bulk composition or is it enriched in Fe? If this solid is not shaped by melt segregation, then I suppose that melt segregation near the CMB is what creates the Fe-rich BMO. So it would be good to illustrate and describe these processes in more detail.

Reply: This is a well-taken point. To first order, segregation in the top boundary layer is responsible for producing the low-pressure chemical fractionation signature but does not directly control the sinking of those solids. Instead, these solids that precipitate at the planet's surface sink due to Rayleigh-Taylor (RT) instabilities, not as settling crystals within a melt. However, at a second order, segregation contributes to this process, as these RT instabilities are driven by both thermal and chemical density contrasts. Segregation dictates the chemical differences between these crystal-rich blobs and the average mantle composition. We have added a new paragraph in the main text (see lines 213-230) and a section in the supplementary material (see Section SI 2.1) to elaborate on the fluid dynamics occurring in the top and bottom boundary layers. In particular, we show that solid-liquid segregation is expected to occur in this thermal boundary layer by comparing the timescale of solid-liquid segregation to the one of the growth of a Rayleigh-Taylor instability.

F. Although I think the hypothesis is important, the manuscript could be much clearer about the detailed scientific logic, and its connection to what occurs in the simulations — particularly the role(s) of melt segregation. Similarly, I would like more clarity on where thermal buoyancy is the key control, and where chemical buoyancy drives things. I think that it is (negative) thermal buoyancy that drives solid from the upper surface to the CMB.

Reply: This is correct.

[...] And I think that melting of these surface-produced solids at the CMB produces an Fe-enriched melt that displaces the residual solid upward [...]

Reply: This is also correct !

[...] but I am not confident that this is what the authors are actually invoking, partly because these processes (particularly the melt segregation) are not illustrated in the manuscript (and partly because the text is not sufficiently clear about this).

Reply: This is something that must be crystal clear for all readers. We have added a paragraph in the main text (see lines 213-230) as well as a new section in the supplementary (see SI Section 2.1). See also our reply to comments D, E and the one below.

Figure 1 shows such a large-scale view of the model that what is happening in the two critical regions (sub-surface and just above the CMB) are barely visible (and there is no indication of the liquid flow field). Moreover, since these seem to be the two crucial, hypothesised processes that BMO formation relies on, they should also be investigated by scaling analysis.

Reply: We acknowledge that the dynamics within the top and bottom thermal boundary layers are complex, making it challenging to clearly interpret the simulation snapshots presented in the manuscript. To address this, we have made several improvements:

- We added a section in the supplementary materials to provide a detailed description of the fluid dynamics in these regions (see SI Section 2.1). In this section, we included a scaling analysis that examines the interplay between the two governing timescales: the timescale of solid-liquid segregation versus the timescale for the development of Rayleigh-Taylor instabilities.
- We added close-up views of the fluid dynamics simulations both near the top thermal boundary layer (Figure S5) and the bottom thermal boundary layer (Figure S6). In this figure, we plotted the velocity fields of both the liquid and solid phase, allowing to better understand what is happening in those regions.
- We introduced figures specifically designed to clarify the dynamics for non specialists in fluid dynamics (Figures S7 and S8).
- Furthermore, we have incorporated a section in the main text that summarizes the key points discussed in the supplementary materials (see lines 213-230).

For the sinking process, I now understand that the transport of solid from the upper, cold TBL to the CMB is driven by thermal (not chemical) buoyancy of a blob that is much, much larger than individual crystals (this was not clear to me on my previous review).

Reply: This is correct.

A key parameter may therefore be the Peclet number, a ratio of a diffusion timescale to an advection timescale. The diffusion timescale is r_0^2/κ , where κ might be adjusted to take account of the latent heat. The advection timescale is H/U , where U is a Stokes settling speed. The blob size r_0 might be approximated as the thickness of the delaminating part of the upper TBL (and its initial temperature and thermal buoyancy from the average temperature of this part). See for example book by Jaupart & Marcheschal. Large Pe corresponds to blobs reaching the CMB.

Reply: Thank you for these suggestions. We have rewritten that section in the supplementary following your recommendations (see SI section 2.3). It is indeed more straightforward that the approach we proposed in the previous version of the manuscript !

Another scaling analysis (or at least detailed consideration) that could/should be included addresses the segregation of melt above the CMB to form a BMO. This would be applied to the solid material that arrives at the CMB, with the question being: does melt have time to segregate (downward) before the bulk material becomes hot enough to rise?

Reply: Thank you for pointing this out. Scaling analysis for this process is now provided in SI section 2.1. This question is basically the symmetric of the processes described in the top boundary layer.

An ambiguous point that should be clearly addressed is the density contrast between liquid and solid. In the shallow mantle, melts are less dense than solids and segregate upward. In the deep mantle and at the CMB, melts are more dense than solids and segregate downwards. There is a density crossover at about 4 GPa, I think. This aspect of the physics would seem important to the clarity of the manuscript but isn't addressed in the main text.

Reply: Indeed, a crucial aspect of the dynamics investigated here is the density crossover between liquid and solid, which occurs at approximately mid-mantle depth, between 40 and 80 GPa, depending on the FeO content in the melt (see Boukaré et al., 2015; Caracas et al., 2019). This crossover is the primary driver for the upward movement of melt in the upper mantle—significantly influencing chemical fractionation in the upper thermal boundary layer—and for the downward movement in the lower mantle, which impacts chemical fractionation and contributes to the formation of the Basal Magma Ocean (BMO) in the bottom thermal boundary layer. To ensure clarity for future readers on the role of this density crossover, we have added several lines to the manuscript (see lines 122-123). This density contrast between solid and liquid plays a major role in the relation between the velocity field of the solid phase and the one of the liquid phase, as shown in Figure S5 and S6.

Here are some other places in the manuscript where I think that the writing could be improved to be clearer about the physics (by line number): 74–78, this sentence seems circular; a magma ocean solidifies from the bottom up because it solidifies from the bottom up. 81–82, this sentence is confusing because it says that efficient surface cooling and top-down solidification leads to bottom-up solidification.

Reply: Thank you for pointing this out. We improved the text.

[...] 151–153, by referring to crystals, this sentence seems to indicate that crystals are settling individually through a melt phase, whereas in fact it is blobs of dense solid that are sinking.

Reply: We have revised the wording, rephrasing "crystal sinking" as "solids sinking as downwellings" where appropriate. See also our reply to point D.

159, this reference to Fe-rich solids confused me because the baseline is unclear (Fe-rich compared to what?). I think that these solids are Fe-rich relative to the residue of melting above the CMB, but it took some thinking to realise/wonder this.

Reply: The term "FeO-rich" adds confusion. We removed it.

187–197, this paragraph might be referring to segregation of the Fe-rich BMO melt from the residual solids, but it isn't very specific. Also, it isn't clear whether "critical" refers to a percolation threshold or the value of a dimensionless number that controls the ratio of two processes.

Reply: We have clarified the text and added several sentences to better connect the main text with supplementary information (see lines 195-204 and 205-212).

242–275, this paragraph discusses both models and observations, I think, but I found it difficult to be certain I understood which was being referred to, particularly in the latter half of the paragraph. Perhaps break into two paragraphs and signpost more clearly where you refer to models versus observations.

Reply: Indeed, we were referencing observations and previous geochemical conclusions in the midst of our model discussion. We have now separated this into two independent paragraphs. First, we present our model (lines 274–285), then we discuss its implications for the early geochemical evolution of the Earth’s mantle (lines 286–303).

References now seem appropriate.

Reply: OK.

See my comments in section F above. I think the lucidity of the manuscript can/should be improved if this is to be published in Nature.

Reply: See our reply to section F. We have been working on this project for two years. Therefore a lot of material that seemed evident to us indeed may have not been obvious to an external reader. The paper was harder to read because important information was missing. Thanks to the thorough review and suggestions, we were able to considerably clarify a number of issues, by adding much needed information, demonstration, regime diagrams, etc., such as those that we addressed in the two rounds of reviews. We believe that the paper reads much more linearly now, and that every step in the deployment of the argumentation is justified. Reading therefore seems more natural, or “lucid” to paraphrase the reviewer.

Reviewer 2

The authors have addressed my previous comments and suggestions. There are of course still many details to be worked out in the long-term, however, I think this paper settles an important open question regarding the formation of basal magma oceans. In particular, freezing from the middle is not a requirement for the formation of such features. This dovetails nicely with other work by some of these authors on an analogous dense layer inside Mars. It may very well be that iron oxides being (1) simultaneously the heaviest major mantle component as well as (2) slightly incompatible with respect to melting causes basal magma/mush oceans to be a ubiquitous feature of all sufficiently oxidized terrestrial bodies that undergo extensive internal melting/freezing.

Reply: Thank you for the positive comment.

Editorial Comments

Data Availability: As a signatory of the Enabling FAIR Data in Earth, space and environmental sciences, Nature is committed to supporting FAIR principles in data sharing and citation. Where community repositories are available, we require data sharing through such repositories for papers in the Earth, space and environmental sciences for papers published in Nature. Where such repositories are not available, datasets may be hosted in generalised data repositories such as Figshare, Dryad or Zenodo. See our FAIR data in Earth science editorial for more details. All of the data and metadata necessary to reproduce your results will accordingly need to be deposited in public DOI-minting repositories, prior to publication, with the DOIs provided in your Data Availability statement. Likewise, please supply a detailed Code Availability statement, describing any restriction of code use.

Reply: We have created a long-term repository where future readers can access the raw files we used to produce the figures in this paper. The link may not be effective yet but it will be accessible in the coming weeks (<https://doi.org/10.18715/IPGP.2024.m42039nd>).

Order: The order in which your text file (without embedded figures) should be presented is as follows: title, authors, authors' affiliations, main text, references from main text, main figure legends, Data and Code Availability statements, acknowledgments and author contribution statements. Please supply the three main figures as separate files in editable format (not flattened), without titles or legends.

Reply: We have added the required Data and Code Availability statements in the manuscript, and followed the order of the manuscript and file format of the figures.

Title: Your present title is somewhat longer than we can accommodate. Would something like "Solidification of Earth's early mantle led inevitably to a basal magma ocean" suffice? Please feel free to suggest an alternative title bearing in mind that it should be 76 characters or less in length (including spaces) and not contain punctuation.

Reply: Yes, this title works ! We also removed the word "early" to remain below the 75 characters limit.

Summary Paragraph: All Nature papers begin with a fully-referenced paragraph, typically no longer than 200 words, aimed at readers in other disciplines. This paragraph starts with a 2- to 3-sentence, basic introduction to the field; continues with a 1-sentence statement of the main findings starting 'Here we show' or an equivalent phrase; and finally, concludes with 2 to 3 sentences putting the main findings into general context so it is clear how the results described in the paper have moved the field forward. A downloadable, annotated example is available at <https://www.nature.com/nature/for-authors/formatting-guide>. In some cases it may be necessary to exceed this limit in order to explain complex material for readers in other fields – in such cases, summary paragraphs can be up to 230 words in length. The extra length, however, is for introduction and context, and not for additional technical information.

Reply: The summary paragraph has been rewritten to follow Nature papers' format and now contains 194 words, which is below the maximum word limit required.

References: Please cite all references as superscripted numbers, rather than in brackets.

Reply: We now used superscripts for all references.

Sub-headings: In our new all-Article format, we strongly encourage several sub-headings within the main text of short Articles to break up the text. Such sub-headings should only be up to 40 characters in length, including spaces. Please also avoid generic sub-headings such as ‘Discussion’ or ‘Conclusions’, and instead indicate the specific content of each section.

Reply: Sub-headings have been added to improve the readability of the manuscript.

Transparent Peer Review: Nature offers a transparent peer review option for new original research manuscripts. We encourage increased transparency in peer review by publishing the reviewer comments and author rebuttal letters if the authors agree. Such peer review material is made available as a supplementary peer review file. Please state in the cover letter ‘I wish to participate in transparent peer review’ if you wish to opt in, or ‘I do not wish to participate in transparent peer review’ if you wish to opt out. Failure to state your preference will result in delays in accepting your manuscript for publication. If you wish to opt in to transparent peer review, please provide your response to reviewers as a .doc file where possible. Please note: we allow redactions to authors’ rebuttal and reviewer comments in the interest of confidentiality. If you are concerned about the release of confidential data, please let us know specifically what information you would like to have removed. Please note that we cannot incorporate redactions for any other reasons. Reviewer names will be published in the peer review files if the reviewer signed the comments to authors, or if reviewers explicitly agree to release their name. For more information, please refer to our FAQ page.

Reply: We are happy to participate in the transparent peer review program and explicitly indicated this in the revised manuscript.

Supplementary Information: Supplementary Information is online-only material published with the manuscript (<https://www.nature.com/nature/for-authors/supp-info>). For most papers, there should be no need for Supplementary Information beyond that already provided as Methods and Extended Data, the aim being to avoid unnecessary fragmentation of the paper online. Exceptions to this rule include large datasets that cannot be accommodated within Extended Data; video material; and more complex “Supplemental Methods” (and any associated references) that do not readily fit within the constraints of the Methods/Extended Data formats. Please note that after the paper has been formally accepted you can only provide amended Supplementary Information files for critical changes to the scientific content, not for style. You should clearly explain what changes have been made if you do resupply any such files.

Reply: We moved the governing equation section to the methods of the main text. We believe that the rest of the supplementary information should remain in the supplement.

ORCID: Nature is committed to improving transparency in authorship. As part of our efforts in this direction, we are now requesting that all authors identified as ‘corresponding author’ create and link their Open Researcher and Contributor Identifier (ORCID) with their account on the Manuscript Tracking System prior to acceptance. ORCID helps the scientific community achieve unambiguous attribution of all scholarly contributions. You can create and link your ORCID from the home page of the Manuscript Tracking System by clicking on ‘Modify my Springer Nature account’ and following the instructions in the link below. If you experience problems in linking your ORCID, please contact the Platform Support Helpdesk.

Reply: The manuscript is now linked to the ORCID number of Boukaré.

Display items: We ask that you take stock of all the data that have been generated throughout the review process and ensure that only the data most central to the conclusions are presented in the main figures. Figures should be comprehensible to readers in other or related disciplines, and assist their understanding of the paper. We encourage authors who are describing complex processes to include a schematic of the main finding as part of the Extended Data to aid readers unfamiliar with the immediate discipline. Figures should be as small and simple as is compatible with clarity. All panels of a figure should be logically connected; each panel of a multipart figure should be sized so that the whole figure can be reduced by the same amount and reproduced on the printed page at the smallest size at which essential details are visible. For guidance, Nature's standard figure sizes are 89 mm (one column), 120 mm (one and a half columns), or exceptionally 183 mm (two columns) wide; the full depth of a Nature page is 247 mm. All panels of figures should be presented on a single page and assembled into a rectangular shape for publication; please indicate any essential alignments (parts horizontal, vertical, spacings of stereo pairs, etc.).

Reply: Our manuscript is composed of three main figures. Ideally, we would like Figure 1 to remain two columns wide. We modified Figure 2 and Figure 3 such that they fit in one column.

Figure formatting: Lettering in all figures (labelling of axes and so on) should be in uniform, sans-serif font, in lower-case type, and large enough to permit substantial reduction for publication (minimum font size 5 pt). Separate parts of a figure are labelled a, b, etc. Units have a single space between the number and the unit, and follow SI nomenclature or the nomenclature common to a particular field. Thousands are separated by commas (1,000). Unusual units or abbreviations are defined in the legend. Scale bars rather than magnification factors should be used.

Reply: We believe our figures follow this format.

Main text statements: We require authors to provide a detailed Author Contribution statement immediately after the acknowledgements; the specific contributions of each author must be listed. It is also a condition of publication that authors include an Author Information statement indicating how to access information regarding reprints and permissions, stating whether or not there is a financial or non-financial competing interest, and naming the author to whom correspondence and requests for materials should be addressed. Please ensure that this section is included in the manuscript file after the Methods (but before the Extended Data legends) - it will not appear in the print version but will appear online in the full-text HTML and PDF versions.

Reply: We have written these statements.

Source Data: To further increase transparency, we encourage authors to provide, in spreadsheet form, the data underlying the graphical representations used in the figures. This is in addition to our well-established data-deposition policy for datasets. Readers of the online manuscript will be able to access the Source Data directly from the figure legend. Spreadsheets can only be submitted in .xls, .xlsx or .csv formats. One file per figure is permitted; thus, if there is a multi-panelled figure the Source Data for each panel should be clearly labeled in the csv/Excel file; alternatively, the data for a figure can be included in multiple, clearly labeled sheets within an Excel file. File sizes of up to 30 MB are permitted; however, it is expected that the vast majority of Source Data files will be considerably smaller than this. When submitting these files with your manuscript, please select the file type "Source Data" and use the title field in the file description tab to indicate the figure to which the Source Data pertain.

Reply: Sources files have been converted in xlsx.

Third party rights: You must provide proof that you have secured permission to use any third party materials that appear in any part of your manuscript, including extended data and supplementary information. Please fill out a <https://www.nature.com/documents/thirdpartyrights-origres.docx>>Third Party Rights Table, and upload this to our manuscript tracking system with the final version of your manuscript. Third party materials include any figures, tables, images, videos or text boxes that are reproductions or adaptations of items that have previously been published elsewhere and/or are owned by a third party. This includes pictures taken by professional photographers, maps and images downloaded from the internet. You will need to obtain the right to use each of these items before your paper can be accepted for publication. You will also need to give proper attribution to the copyright holders in your paper. Please ensure you upload any necessary grants of rights alongside the final version of your manuscript. More information is available on our <https://www.springernature.com/gp/partners/rights-permissions-third-party-distribution>>Rights and permissions page. Failure to obtain the appropriate rights and to supply a completed third party rights table will delay the publication of your article.

Reply: We filled out the required form. No third party material was used for this manuscript.

Cover artwork: We welcome submissions of artwork for consideration for our cover. More information can be found in our guide for cover artwork. The file name(s) should include the manuscript reference number and be labelled as a cover suggestion; a short description is also preferred. Illustrations should be selected more for their aesthetic appeal than for their scientific content. We cannot promise that your suggestions will be selected for the cover, as competition is intense.

Reply: If the paper is accepted, we will work on a cover artwork.

Required Files Checklist

A cover letter describing your response to any editorial comments and detailing any format changes during revision, particularly if the overall length is affected.

Reply: Done.

A separate point-by-point response to the remaining issue raised by referee 1.

Reply: Done.

The final version of your text as a Word document (Word Equation Editor/MathType should be used only for formulae that cannot be produced using normal text or Symbol font). If this is not possible, please provide the manuscript as a single plain vanilla TeX or LaTeX file that includes all references and abbreviations, with no special formatting, as well as a PDF version that is uploaded as a 'related manuscript file'.

Reply: Our manuscript has been uploaded in two versions : 1) one unique TeX file that includes the .bbl and 2) a pdf file.

Production-quality versions of the three main figures (for details see <https://tinyurl.com/yaajfusxw>). As we need to be able to edit the figures so that they conform to our house style, the submission of files that are incorrectly formatted, flattened, or of insufficient resolution may delay final acceptance of your manuscript.

Reply: OK.

The Source Data, as a separate table file for each figure for which it is provided.

Reply: OK.

The final version of your Supplementary Information files (as a single PDF file and two video files). Please also supply a separate SIGuide text file, with a short description of the contents of the Supplementary Information files (including titles and legends to accompany the videos).

Reply: OK.

For optimal quality videos please use a H.264 encoding and the standard aspect ratio of 16:9 (4:3 is second best), and do not compress the video.

Reply: OK.

Completed and signed copy of the manuscript checklist (<https://www.nature.com/documents/nature-manuscript-checklist-research.pdf>). NOTE: This form should be uploaded as scanned PDFs as a separate attachment, by choosing the file type 'Related Manuscript File', here is the link.

Reply: OK.

Completed copy of the editorial policy checklist (<https://www.nature.com/documents/nr-editorial-policy-checklist.pdf>). NOTE: This form should be uploaded as scanned PDFs as a separate attachment, by choosing the file type 'Related Manuscript File', here is the link

Reply: OK.

Completed and signed copy of the colour charge form (<https://www.nature.com/documents/nature-colour-figure-form.pdf>). Nature requests that authors of accepted manuscripts contribute towards the total cost of reproduction of colour figures in print. NOTE: This form should be uploaded as scanned PDFs as a separate attachment, by choosing the file type 'Related Manuscript File', here is the link

Reply: OK.

Reviewer 1

The manuscript is improved after the latest round of revision. I think that the idea is novel and important, in that it builds a robust physical explanation for the existence of a basal magma ocean from the dynamics of mantle convection, magma segregation, chemical fractionation, and phase change, without relying on the details of the liquidus and geotherm curves.

Reply: Thank you. We appreciate this positive feedback.

*The writing could be improved to bring the ideas into sharper focus. I think the authors could do more to provide a simple explanation for where and why chemical fractionation (i.e., Fe enrichment) occurs and how that material ends up in a basal magma ocean. For example, the abstract mentions the importance of melt segregation but doesn't note the importance of this process in the *shallow* mantle, which is the most innovative part of this paper. I attach an annotated manuscript version with some suggestions.*

Reply: We modified the abstract to include a discussion on the importance of shallow solid-liquid fractionation. We also added a few sentences to clarify the processes that lead to basal magma ocean formation (lines 132-136). We better explained what composition is carried by the crystals forming at shallow depths and how it would affect the composition of the BMO (lines 161-177). Thank you very much for detailed annotated pdf. We have accounted for all your comments, and fixed all the typos you pointed out.

The scaling analyses provided in the supplementary material have evolved to be more convincing in terms of their relevance, but they still seem to provide only marginal support for the conclusions. Nonetheless, I think this is adequate given the uncertainties of the problem and general plausibility of the explanation. It might be worth spending time to tighten up the writing in the supplement too.

Reply: Thank you for your support and for recognizing that our approach remains acceptable given the uncertainties. We further clarified the supplement.

The authors might consider the relevance of the following to their supplementary information on model development: <https://doi.org/10.1093/gji/ggac309>, <https://doi.org/10.1146/annurev-earth-032320-083704>, <https://press.princeton.edu/books/hardcover/9780691176567/the-dynamics-of-partially-molten-rock>

Reply: Thank you for the suggestion. We added these references. We agree that these will help future readers to connect our work to past and ongoing development in the field.

Version 2:

Referee #1
(Attachment):

The solidification of Earth's early mantle leads inevitably to a basal magma ocean

Charles-Édouard Boukaré^{1*}, James Badro^{1†} and Henri Samuel^{1†}

^{1*}Université Paris Cité, Institut de Physique du Globe de Paris, CNRS, 1 rue Jussieu, 75005 Paris, France.

*Corresponding author(s). E-mail(s): boukare@ipgp.fr;

Contributing authors: samuel@ipgp.fr; badro@ipgp.fr;

[†]These authors contributed equally to this work.

Abstract

Current interpretation of ancient geochemical heterogeneities and deep-rooted geophysical structures in the mantle is that they stem from the solidification of Earth's primitive magma ocean. This requires that as contrary to the Moon, Earth's magma ocean has crystallised downwards, producing a basal magma ocean above the core. This hypothesis remains debated ~~x~~ on the basis of whether the first solids form at the bottom of the mantle giving rise to upward solidification to the surface, or whether solids form above the melts and solidify downwards to the core-mantle boundary. We numerically model ~~x~~ the mantle's solidification using a novel multiphase fluid dynamics approach integrated with melting phase diagrams and phase relations. We ~~found~~ *find* that gravitational segregation of dense, iron-rich melts from lighter, iron-poor solids is the main engine of dynamical evolution, irrespective ~~x~~ of where melting curves and geotherms intersect in the mantle. The outcome is an unavoidable production and accumulation of melt above the core, giving rise to a basal magma ocean. Our fluid dynamics methodology is coupled with petrological and geochemical models that allows estimating the compositional signature and spatial distribution of primordial reservoirs and heterogeneities. We find that these can be preserved throughout magma ocean solidification and provide a robust petrological and geodynamical rationale for a magma-ocean origin of the aforementioned geophysical and geochemical structures.

where?

047 **Keywords:** Mantle Structure and Dynamics, Mantle Differentiation, Magma
 048 Oceans, Rocky Planets Formation and Evolution.

049
 050

051

052 ~~The observation of~~ isotopic anomalies in short-lived radiogenic isotopic systems in mantle rocks ~~that~~ record magmatic differentiation processes occurring in the first 100 Myrs ~~show~~ that the Earth's mantle has preserved chemical heterogeneities [1–5] dating back to its infancy. These findings are corroborated by the noble-gas geochemical record that ~~also~~ argues for the preservation of such early formed geochemical reservoirs [6–8]. The solidification of a deep, primitive magma ocean alone can explain such an early silicate differentiation event [2–4], as it produces the observed fractionation in several trace-element ratios at a global scale. This solidification process can also explain the current seismic structure of the deep mantle, where Large Low Shear Velocity Provinces (LLSVPs) and Ultra-Low velocity zones (ULVZs) can be interpreted as residual products of primordial magma-ocean solidification [9–13]. As remnants of magma ocean solidification, the two antipodal, deep-rooted (above the core-mantle boundary) LLSVPs [14] must play a first order role on global mantle and core dynamics [15], plate tectonics and hot-spot magmatism [16–19] during the entire history of the Earth. Therefore, understanding magma-ocean solidification from a dynamical and petrological point of view is essential for our comprehension of the long-term evolution of Earth's mantle and its present-day state.

071 These geochemical and seismological observations ^{indicate} ~~require~~ that the last remnants of the terrestrial magma ocean were located deep in the mantle, above the core-mantle boundary, ^{but this} ~~which~~ remains debated both dynamically and petrologically [10, 20–27]. Classical magma ocean solidification models, akin to those developed for the Moon, stipulate ~~solidification from the bottom (core-mantle boundary) upwards~~ [24, 28], ~~and are based on the fact~~ that in a cooling magma ocean, the first solids appear at the bottom of the mantle due to the intersection of liquidus and adiabat at depths, pushing the residual melt upwards ~~–~~ [24, 28]. As crystallization proceeds, solidification is expected to occur from the bottom (core mantle boundary) upwards. An alternative scenario, based on the fact that the solids are more buoyant than the melt in the deep mantle, argues that solidification takes place in the middle of the magma ocean, separating it in basal and shallow magma oceans [10, 23, 26, 29]. The shallow magma ocean solidifies upwards ~~(i.e. classically)~~ quickly because of efficient cooling at the surface, while the basal magma ocean solidifies slowly, pushing the residual melt down to the core-mantle boundary.

087 The issue of top-down *vs.* bottom-up solidification is ultimately controlled by thermodynamic properties ^{that} ~~which~~ determine: 1) where solidification takes place, *i.e.*, intersection of liquidus with temperature, and 2) where solids and liquids accumulate, *i.e.*, the density contrast between melt and solids. In addition, the efficiency of solid-liquid phase separation plays a fundamental role because ~~regardless of the petrological or geochemical nature of solids and~~

092

leading?

~~melts, as~~ no fractionation can occur if melts cannot gravitationally segregate from solids, regardless of the petrological or geochemical nature of solids and melts. Depending on the efficiency of solid-liquid phase separation [30], the magma ocean can freeze as a homogeneous silicate reservoir (batch crystallisation scenario) or form strongly fractionated reservoirs of distinct compositions (fractional crystallisation scenario) [21, 31].

Previous fluid-dynamics studies mostly focused on the issue of solid-liquid phase separation in magma oceans at high melt fraction [21, 31–38]. The question of whether crystals settle or are suspended by the flow have been meticulously investigated. However, in thermal evolution models, crystal fraction is not explicitly transported. ^{In such models,} if crystals can settle, they are deposited at the bottom of the magma ocean at a rate that is a function of the settling velocity corrected by a re-entrainment flux [33, 39]. Alternatively, crystal fraction can be determined assuming local thermodynamic equilibrium by comparing the temperature to the solidus and liquidus at a given depth [e.g., 36, 40].

Except for a few works [e.g., 41], less attention has been given to the later stages of magma ocean differentiation, where the mantle enters a solid-like dynamic regime due to high crystal fraction. While previous fluid-dynamics simulations have investigated the issue of phase separation efficiency, they did not account for transport of solids and liquids of distinct composition, crystal production, and remelting. These assumptions prevent the possibility of scenarios where (1) a clear distinction can be made between the location of crystal formation and that of crystal accumulation, and (2) radial magma ocean structure is controlled by the location where crystals settle, rather than the location where they form.

To address these challenges, we use a novel fluid-geodynamical approach with the numerical code **Bambari** (SI Section 1.1 to 1.5). **Bambari** implements a multiphase flow mathematical formalism based on the averaging method [30, 42–48]. We systematically explored a ~~full~~ range of mantle differentiation scenarios, ~~through~~ ~~two~~ independent mechanisms: petrological differentiation (chemical partitioning) and mechanical differentiation (solid-liquid phase separation). Our simulations account for a density crossover between melt and solid at mid-mantle depth (SI Section 1) [25, 26, 49]. We focus on magma ocean solidification dynamics beyond the rheological transition, *i.e.*, when the mantle starts to behave as a solid. ↪ jargon

We ^{find} ~~found~~ that an iron-rich Basal Magma Ocean (BMO) always forms above the core-mantle boundary (Figure 1), and that it corresponds to the last residue of mantle solidification. This finding is irrespective of the depth where liquidus and adiabat intersect [23, 24] (*i.e.*, where solidification is expected to occur). We ~~shall~~ first consider the least favourable scenario to produce a BMO [24], when the adiabat intersects the liquidus at the bottom of the mantle, ^(and radial models expect solidification to take place there and proceed upwards.)

The initial conditions of our simulations correspond to a compositionally homogeneous mantle at 50% melt fraction. Large solid production ~~is seen~~ at

homogeneously distributed

occurs

awk

139 the surface in the earliest stage of evolution (Fig. 1a-b) because temperature
 140 in the shallow mantle drops below the solidus, regardless of where liquidus and
 141 adiabat curves intersect at depth. ~~The newly formed solids are denser than~~
 142 ~~melts, and sink~~ This crystal-rich shallow layer becomes rapidly gravitationally
 143 unstable both thermally - because it is colder than the underlying interior -
 144 and compositionally - because it is composed of negatively buoyant crystals
 145 (see SI Section 2.1) forming a Rayleigh-Taylor instabilities. This crystal-rich *awk*
 146 unstable layer then sinks as cold downwellings. However, ~~they~~ the solids do
 147 not accumulate in the deep mantle because they progressively remelt during
 148 their descent (Fig. 1a-c).

* 149 With further cooling of the magma ocean (Fig. 1d-f), solids formed at shall- *→ What*
 150 low depths gradually accumulate in the lower mantle. Our simulations show *chemical*
 151 that these solids, which are still fed from the surface by cold downwellings, do *signature*
 152 not remelt anymore (Fig. 1d-f). Notably, these deep-seated solids are not the *do they*
 153 result of the crystallization of deep mantle melt (where the liquidus intersects *carry?*
 154 the adiabat). This key observation is at odds with all previous assessments *Fe-rich?*
 155 based on 1D solidification models, ~~showcasing~~ *high lightning*
 156 variations in temperature and composition that cannot be accounted for in 1D
 157 models. The reason why solids do not significantly remelt during their descent
 158 is that (1) the average temperature in the mantle (once the rheological transi-
 159 tion is reached) is lower than the liquidus at all depths, ~~and~~ (2) the slope of the
 160 liquidus favors crystallization at larger depths. ~~Consequently, thermal diffusion~~
 161 ~~remains the only process capable of remelting crystals during their descent in~~
 162 ~~the mantle, and timescale arguments indicate that crystals formed at shallow~~
 163 ~~depths sink faster than they can, and~~ (3) the crystal-rich downwellings sink
 164 too fast to heat up and remelt by thermal diffusion (see SI Section 4.6.2.3).

165 The final stage involves the accumulation of denser solids sinking ~~from the~~
 166 ~~top as downwellings~~ in the deep mantle (Fig. 1e-f). This forms a thermal lid
 167 above the core, leading to efficient reheating of the lowermost mantle. This
 168 in turn melts mantle fusible and FeO-rich components, that are so negatively
 169 buoyant that they accumulate at the base of the mantle, forming a basal
 170 magma ocean (Fig. 1g-h). This ~~ongoing process of feeding iron-rich~~ continuous
 171 transport of solids from the surface to the lower mantle, where ~~it remelts~~
 172 ~~to segregate gravitationally they remelt and settle~~ above the CMB ~~effectively~~
 173 ~~enriches it in iron with respect, gradually enriches the BMO in FeO compared~~
 174 to the Bulk Silicate Earth (BSE) ☹️

175 Let us now discuss the case ~~[23]~~ of a steeper liquidus curve [23] intersecting
 176 the adiabat at mid-mantle depths, where a basal magma ocean is the expected,
 177 final stage of mantle solidification. We find that the dynamical evolution is
 178 essentially ~~similar~~ *the same* (see SI Fig. S10S18), which ~~further~~ *?*
 179 the crossing of the liquidus and adiabat does not play a major role in the style of
 180 solidification of the magma ocean. A liquidus curve that intersects the adiabat
 181 in the mid-mantle facilitates remelting in the lower mantle, promoting the
 182 formation of a basal magma ocean. Even though the crystal fraction is higher
 183 in the mid-mantle, this sluggish shell does not isolate the upper mantle from
 184

The solidification of Earth's early mantle leads inevitably to a basal magma ocean

the lower mantle. Downwellings that form at the surface do not accumulate at mid-mantle depth as they remain thermally negatively buoyant (SI Section 2, Fig. S8). They pursue their descent and remelt in the deep mantle.

We conclude that the location where solids are first produced is of secondary importance compared to the location where solids and liquids of distinct composition eventually accumulate. This depth of accumulation is primarily controlled by the density contrast (~~the buoyancy~~) between liquid and coexisting solid silicates (*i.e.*, solids in thermodynamic equilibrium with that melt), which is the main driving force of magma-ocean dynamics on Earth. At the end of magma ocean solidification, the resulting iron-rich thermochemical structures in the lowermost mantle (Fig. 1) are geophysically and mineralogically consistent with the properties of LLSVPs (Fig. 1) and ULVZs [9–13, 50, 51]. ~~This puts a magma ocean origin of such primordial heterogeneous reservoirs on strong grounds.~~

The solidification sequence described above requires the solid-liquid phase separation to be faster than mantle solidification and re-entrainment remixing by thermochemical convection. Phase separation efficiency is controlled by the dimensionless melt mobility number, δ (SI Section 1.3, 2.2, Fig. S5a, and Fig. S7S11), which is mainly governed by crystal size and melt viscosity.

In our simulations, while the melt is several orders of magnitude more viscous than magma ocean's values, we preserve a realistic balance of the thermochemical convection velocity and phase separation velocity - and thus the extent of chemical differentiation - by using a relatively large melt mobility number (see SI Section 2.2). To validate our approach, we explored numerically the $Ra\delta$ parameter space that captures the competition between chemical differentiation and convective mixing in a dimensionless way. For each thermal Rayleigh number we investigated, we were able to identify a critical melt mobility number above which substantial chemical differentiation is observed (see SI Section 2.2).

By extrapolating our regime diagram to magma ocean conditions, assuming a liquid silicate viscosity of 1 Pa s [52, 53], melts ~~will~~ segregate from the solids if the crystal size is larger than 0.01 μm . Even assuming a four-orders-of-magnitude higher viscosity (10^4 Pa s) the critical crystal size above which melt segregation can occur is 1 μm (SI Section 1.5.2.2), consistent with previous estimates [21, 22, 31, 54]. Because grain sizes larger than 1 mm are expected in geophysical contexts, phase separation will always be efficient and dominate re-entrainment remixing in realistic conditions.

A fundamental aspect is that the fluid dynamics within or in the vicinity of the top and bottom thermal boundary layer is at the heart of the chemical evolution of the magma ocean. Crystals formed at the surface of the planet experience solid-liquid chemical fractionation governed by low-pressure mineral phases. To transport a fractionated composition into the deep mantle, solids and liquids must segregate from each other in the vicinity of the cold thermal boundary (SI Figures S7 and S9). Otherwise, it is the bulk composition of the liquid that is brought to the deep mantle. Scaling analysis shows that the

But shallows-formed solids are Fe-rich...?

?

Define

?

vague

?

185
186
187
188
189
190
191
192
193
194
195
196
197
198
199
200
201
202
203
204
205
206
207
208
209
210
211
212
213
214
215
216
217
218
219
220
221
222
223
224
225
226
227
228
229
230

231 solid-liquid segregation in this layer is expected to be faster than the growth
 232 rate of Rayleigh-Taylor instabilities (SI section 2.1) in the Earth's magma
 233 ocean, allowing for chemical shallow differentiation to occur. Although present,
 234 scaling analysis indicates that this process is in fact less pronounced in our fluid
 235 dynamics simulations than it would be under real conditions. The symmetric
 236 version of the processes described above is responsible for the formation of the
 237 BMO. FeO-rich melt migrates downwards within the hot thermal boundary
 238 layer before being entrained in upwelling currents (SI Figures S8 and S10).

239 We explore the geochemical consequences of our dynamical model on
 240 the production and nature of primordial mantle heterogeneities, inherited
 241 from magma ocean solidification. We use experimentally-determined parti-
 242 tion coefficients of trace elements between melts and the liquidus phase at
 243 upper mantle (olivine) [55] and lower mantle (bridgmanite) [56, 57] conditions
 244 to track a simple (no garnet and no ferropericase) evolutionary model of key
 245 trace-elements elemental ratios (Sm/Nd, Lu/Hf, Hf/W) during solidification
 246 (SI Section 1.4). At the end of mantle solidification, solids in the upper man-
 247 tle show superchondritic Lu/Hf ratios (in yellow to red, in Fig. 2a), due to
 248 olivine crystallisation during bottom-up crystallisation of the upper mantle.
 249 Conversely, in the lower mantle above the BMO, we observe subchondritic
 250 Lu/Hf ratios (in blue to white, in Fig. 2) that indicates bridgmanite fraction-
 251 ation. Furthermore, we observe that solidification and remelting at different
 252 depths produce a complex (marble cake-like) geochemical structure in the solid
 253 mantle from core to crust (Fig. 2), far from a vision inspired by geochemical
 254 two-box modelling of single enriched (residual liquid) and depleted (precipitat-
 255 ing solids) reservoirs [56, 58]. While the extent and magnitude of trace-element
 256 ratio distribution depends dramatically on phase separation efficiency (Fig. S6
 257 and S7), planetary scale solidification systematically generates heterogeneities
 258 at all depths (Fig. 2).

259 The Lu/Hf ratio is a suitable geochemical tracer to quantify the extent of
 260 low pressure (*i.e.*, olivine) mantle solidification on a global magma ocean [59].
 261 This is because Lu is incompatible (*i.e.*, enriched in the melt) both in olivine
 262 and bridgmanite, while Hf is incompatible (*i.e.*, enriched in olivine, but compatible in bridg-
 263 manite. In this simple model, solids with superchondritic (high) Lu/Hf ratios
 264 stem from olivine crystallisation and denote a shallow origin, while solids with
 265 subchondritic (low) Lu/Hf ratios originate from bridgmanite crystallisation
 266 and can only have formed at depth. The extent of mixing between these two
 267 components on the planetary scale can be seen on a Sm/Nd-Lu/Hf correla-
 268 tion map (Fig. S8). One should note however that the extent and amplitude
 269 of fractionation shown in Fig. 2 varies with the efficiency of solid-liquid phase
 270 separation (see Fig. S6 and S7 for more extreme cases). The amount of stirring
 271 obtained in these 2D Cartesian geometry may also be quantitatively affected
 272 by three-dimensional effects and sphericity. Unlike the lunar magma ocean,
 273 the Earth's magma ocean does not evolve as a stack of immobile layers each
 274 retaining the geochemical characteristics of the depths where they formed.
 275 Rather, primordial heterogeneities are scattered at all depths. This marks a
 276

shift in our understanding: the way lunar-like magma oceans evolve, based on the differentiation observed in magma chambers on Earth, does not necessarily apply to mantles as deep as the ones of the Earth or Mars [60]. There, despite stirring during solidification caused by thermo-chemical convection, early-formed geochemical reservoirs are preserved and scattered throughout the mantle [61–63].

Finally, we analysed the statistical distribution of geochemical heterogeneities in the solid and molten (BMO) lower mantle, and more specifically the Lu/Hf population statistics (Fig. 3). If we only account for bridgmanite fractionation (Fig. 3a) and disregard olivine fractionation in the upper mantle, then the distribution of Lu/Hf heterogeneities is centered around 0 for the solid mantle, and the BMO composition is shifted towards higher Lu/Hf ratios (70% higher than the BSE) as one would expect from bridgmanite-melt fractionation. However, when both olivine-melt and bridgmanite-melt fractionations are accounted for (Fig. 3b), the distribution of Lu/Hf heterogeneities for the solid mantle and the BMO are both centered around 0, showing that the opposite effects of low and high pressure solid-melt fractionation cancel each other out. ~~This implies that the composition of the Primitive Upper Mantle (PUM) integrates the signature of a complex mixture of shallow and deep geochemical components (Fig. 3), both spatially and temporally. Concomitantly, the signature of olivine-melt trace elements fractionation is present in the lower mantle at the end of magma ocean solidification and can be preserved over geological times. This~~

The role of low-pressure chemical fractionation on deep mantle composition has a fundamental consequence ~~for~~ the quantification of the extent of magma ocean solidification differentiation from the standpoint of trace element ratios in mantle rocks. The ~~PUM is unfractionated in refractory lithophile elements~~ primitive upper mantle (PUM) has unfractionated refractory lithophile trace element ratios [64], and because bridgmanite crystallisation strongly fractionates some of these ratios [56], this has been used to constrain the ~~extent~~ maximum amount of bridgmanite-melt differentiation that ~~happened—took place~~ in the early Earth. On this basis, and in order to not disturb those ratios in the PUM (within their uncertainties), no more than 8% [56, 57, 65] bridgmanite can be crystallised alone in the magma ocean, which is inconsistent with experimental melt relations and the melting phase diagrams [66] of pyrolite. Our work relaxes this constraint, because bridgmanite-melt partitioning is not the only relevant process that describes the chemical differentiation between the PUM and the lower mantle during magma ocean solidification. Olivine-melt partitioning plays an important role as well, because cold downwellings form at the surface of the planet, constantly feeding the lower mantle with a PUM-like signature. These two effects effectively smooth each other out) allowing for a much larger extent of lower-mantle silicate differentiation (*i.e.*, bridgmanite-melt) to take place without affecting chondritic ratios in the PUM (see Fig. 3).

→ have signatures that partially cancel each other.

8 *The solidification of Earth's early mantle leads inevitably to a basal magma ocean*

323 We Finally, we conducted simulations to quantify the outgassing of volatile
 324 species in our simulations (see SI Section 2.23.2). We used a Lagrangian
 325 approach to track and to estimate the amount of mantle materials that would
 326 reach the exsolution depth. We assumed that if mantle parcels do not reach
 327 the exsolution depth, they remain undegassed, and retain their primordial
 328 noble gas signature and volatile content. Consistent with the work in [67],
 329 our results indicate that only a negligible fraction of the mantle is expected
 330 to degas during magma ocean solidification beyond the rheological transition.
 331 This would be consistent with the noble gas geochemical record that suggests
 332 the preservation of an early-formed geochemical reservoirs [6–8] although this
 333 topic remains highly debated [68, 69].

334 Our modelling was performed using the least favourable conditions to pro-
 335 duce a BMO, from both a geochemical (full equilibrium fractionation, see
 336 SI Section 1.4.1) and a petrological (adiabat intersects the liquidus at the
 337 base of the mantle) standpoint; yet, (1) it systematically produces a basal
 338 magma ocean in the final stage of evolution, which appears to be inevitable on
 339 Earth, and extensible to other large terrestrial planets with Earth-like com-
 340 position and (2) the geochemical imprint of this solidification on the solid
 341 mantle is far less marked than that predicted by two-box geochemical models
 342 [59], due to extensive vertical mixing during solidification. The composition of
 343 the PUM integrates the signature of a complex mixture of shallow and deep
 344 geochemical components. Concomitantly, the signature of olivine-melt trace
 345 element fractionation is present in the lower mantle at the end of magma
 346 ocean solidification and could be preserved over geological times.

347 These findings call for a re-interpretation of the available geochemical
 348 record and geophysical observations to better reconstruct the Earth's thermal
 349 and chemical history from its infancy to the present-day, and more generally
 350 to better understand the diversity of terrestrial bodies.

351
 352
 353
 354
 355
 356
 357
 358
 359
 360
 361
 362
 363
 364
 365
 366
 367
 368

solid-melt phase separation. Convective motion in a solidifying magma ocean is driven by three types of density differences. These originate from thermal expansivity (due to temperature differences), compositional differences (due to changes in iron content) and phase changes (due to the varying melt fraction). An additional mechanism that generates motion in a partially molten convective medium is the solid-liquid phase separation driven by shear deformation, in addition to the density contrasts between the melt and the solid. Phase separation is limited by matrix deformation, *i.e.*, compaction, and viscous friction between the melt and the solid via Darcy law. We implemented a depth-dependent density contrast between liquid and solids [16–18], and used a thermodynamically self-consistent compositional evolution of melts and solids based on recent experimental melting phase diagrams [16, 19] (Section 1.4, Fig. S1 and S2).

The model is composed of two mechanical phases: a liquid phase and a solid phase. Each mechanical phase is composed of two major compositional phases that correspond to a FeO-rich and member and a MgO-rich end-member. Trace elements are distributed among the mechanical phases according to their solid-liquid partitioning coefficients. In addition to the advection-diffusion of temperature, our model tracks the advection of 14 compositional fields: 7 liquids (active MgO-rich end-member, FeO-rich end-member, and passive chemical species Hf, Sm, Nd, W, Lu) and the 7 solids counterparts.

In the following, we recall the governing equations solved by the code in their dimensionless form. We non-dimensionalize lengths by the thickness of the mantle H , time by the thermal diffusion scale H^2/κ (κ being the the average coefficient of thermal diffusivity), velocities by κ/H , temperatures by the total super-adiabatic temperature contrast across the upper and lower thermal boundary layers in the mantle ΔT_m and pressure and deviatoric stress using viscous pressure scale $\eta_s \kappa/H^2$.

1.1 Mass conservation equations

The mass conservation equations for the four major chemical components write:

$$\begin{aligned}
 \frac{\partial \phi_1}{\partial t} + \nabla \cdot \mathbf{v}_l \phi_1 &= \Gamma_1 \\
 \frac{\partial \phi_2}{\partial t} + \nabla \cdot \mathbf{v}_l \phi_2 &= \Gamma_2 \\
 \frac{\partial \phi_3}{\partial t} + \nabla \cdot \mathbf{v}_s \phi_3 &= -\Gamma_1 \\
 \frac{\partial \phi_4}{\partial t} + \nabla \cdot \mathbf{v}_s \phi_4 &= -\Gamma_2
 \end{aligned} \tag{1}$$

where \mathbf{v}_l is the velocity of the liquid phase, \mathbf{v}_s is the velocity of the solid phase and Γ_i are associated with the rate of phase change. The subscripts 1 and 3

cite
rusok
et al
2022

drag

REFs

Fig. S1 Density contrast parameterization. (a) Density of solid and liquid silicate as function of pressure for various composition. (b) Density difference between solid and liquid. The thick grey lines correspond to the exact difference between the density curves shown on the left. The color lines depict the parameterization we used in this study. The iso-chemical density contrast is shown in blue ($\Delta\rho_1$ in Eq. (2)) and capture to the effect of highest compressibility of silicate melt compared to their solid counterparts. The chemical density contrast correspond to the density contrast between iron-rich and iron-free silicates ($\Delta\rho_2$ in Eq. (2)).

refer to the MgO end-members in the solid and liquid phase, respectively. The subscripts 2 and 4 refer to the FeO end-members in the solid and liquid phase, respectively. The densities of the four chemical components are:

$$\begin{aligned}\rho_1 &= \rho_0(1 - \alpha T) + \frac{1}{2}\Delta\rho_1 - \frac{1}{2}\Delta\rho_2 \\ \rho_2 &= \rho_0(1 - \alpha T) + \frac{1}{2}\Delta\rho_1 + \frac{1}{2}\Delta\rho_2 \\ \rho_3 &= \rho_0(1 - \alpha T) - \frac{1}{2}\Delta\rho_1 - \frac{1}{2}\Delta\rho_2 \\ \rho_4 &= \rho_0(1 - \alpha T) - \frac{1}{2}\Delta\rho_1 + \frac{1}{2}\Delta\rho_2\end{aligned}$$

normally phase's have densities, not chemical components.

where ρ_0 is the reference density, the thermal expansivity α is considered constant in the solid and liquid phase and T is the temperature. The isochemical density contrast between the liquid and solid, is equal to $\Delta\rho_1 = \rho_1 - \rho_3 = \rho_2 - \rho_4$. Similarly, the chemical density contrast between the dense and light end-members is $\Delta\rho_2 = \rho_2 - \rho_1 = \rho_4 - \rho_3$. We parameterized these density contrasts to fit a self-consistent thermodynamic model [16] (Fig. S1).

1.2 Energy conservation equation

We assume local thermal equilibrium between all the phases, *i.e.*, the temperature at a given location in all the phases are the same ($T_1(x, z, t) = T_2(x, z, t) = T_3(x, z, t) = T_4(x, z, t)$). This leads to a single equation for the conservation of energy:

$$\partial_t T + \bar{\mathbf{v}} \cdot \nabla T = \nabla^2 T + (\Gamma_1 + \Gamma_2) S_t, \quad (3)$$

where $\bar{\mathbf{v}}$ is the velocity of the solid-liquid mixture, S_t is the dimensionless Stefan number that writes,

$$S_t = \frac{L}{C_p \Delta T_m}, \quad (4)$$

where C_p is the thermal capacity of the mixture of the four components, and L is the latent heat release during phase change.

The velocity of the solid-liquid mixture is related to the solid and liquid velocities via the following equation:

$$\bar{\mathbf{v}} = \phi \mathbf{v}_s + (1 - \phi) \mathbf{v}_l. \quad (5)$$

1.3 Momentum conservation equations

Since we account for two mechanical phases, two momentum conservation equations are required to describe the two-phase flow mechanical interactions. The following Stokes equation describes the momentum conservation for the averaged solid-liquid mixture:

$$-\nabla \Pi + \nabla \cdot \bar{\boldsymbol{\tau}} - Ra T \mathbf{g} - Rp \phi \mathbf{g} + \frac{1}{2} Rc (\phi_2 + \phi_4 - \phi_1 - \phi_3) = 0, \quad (6)$$

where $\nabla \Pi$ is the dynamic pressure gradient ($\nabla \Pi = \nabla P - \rho \mathbf{g}$, where P is the total pressure), $\bar{\boldsymbol{\tau}}$ is the deviatoric stress tensor for the solid-liquid mixture, the thermal Rayleigh number, Ra , and the phase and compositional buoyancy numbers, Rp and Rc are respectively:

$$\begin{aligned} Ra &= \frac{\rho_0 \alpha \Delta T_m g H^3}{\kappa \eta_s}, \\ Rp &= \frac{\Delta \rho_1 g H^3}{\kappa \eta_s}, \\ Rc &= \frac{\Delta \rho_2 g H^3}{\kappa \eta_s}, \end{aligned} \quad (7)$$

where \mathbf{g} is the gravity constant. We define the stress tensor of the solid-liquid mixture:

$$\bar{\boldsymbol{\tau}} = (1 - \phi) \boldsymbol{\tau}_s + \phi \boldsymbol{\tau}_l = \eta(\phi) \left(\nabla \bar{\mathbf{v}} + [\nabla \bar{\mathbf{v}}]^t \right), \quad (8)$$

SI The solidification of Earth's early mantle leads inevitably to a basal magma ocean

where $\eta(\phi)$ is the viscosity of fluid mixture that varies from $\eta(\phi = 0) = \eta_s$ to $\eta(\phi = 1) = \eta_l$. We use,

$$\eta(\phi) = \eta_s \times 10^{\left(1 + \tanh\left(\frac{\phi - 0.5}{0.1}\right)\right) \frac{\eta_l}{2\eta_s}}. \quad (9)$$

REF?

For the second momentum conservation equation, we use a retro-action relationship (see [1, 10]) that describes $\Delta \mathbf{v} = \mathbf{v}_l - \mathbf{v}_s$, the velocity difference between the two phases:

$$\phi \Delta \mathbf{v} = -\phi^2 \delta^2 \left(\zeta (1 - \phi) \nabla \left[\frac{(1 - \phi)}{\phi} \nabla \cdot \phi \Delta \mathbf{v} \right] + X \mathbf{g} + (1 - \phi) \nabla \cdot \bar{\boldsymbol{\tau}} \right) \quad (10)$$

where

$$X = (1 - \phi) Rp + \frac{1 - \phi}{2} \left(\frac{\phi_3 - \phi_4}{\phi} + \frac{\phi_2 - \phi_1}{1 - \phi} \right) Rc, \quad (11)$$

δ is the melt mobility number, and ζ is the dimensionless compaction viscosity. The melt mobility number and dimensionless compaction viscosity writes,

$$\delta = \sqrt{\frac{r_0^2}{C_0 H^2} \frac{\eta_s}{\eta_l}}, \quad (12)$$

$$\zeta = \frac{\eta_c}{\eta_s},$$

with a the crystal size, C_0 is a constant, and η_c is a constant associated to the compaction viscosity, sometimes called bulk viscosity, that describes the viscous resistance of a solid matrix to compact (*i.e.*, close the porous space) [6, 10, 20, 21]. Here, we follow [10] where the compaction viscosity, η_{bulk} , is defined as $\eta_{\text{bulk}} = \eta_c / \phi$ (the compaction viscosity must be infinite when the melt fraction is 0).

The above equations are discretized using conservative finite-difference schemes. The corresponding set of Equations (6) and (10) involves the inversion of two sparse matrices, which are the most time-consuming operations. In **Bambari**, these matrix inversions are performed using the direct **PARDISO** library that is parallelized with **OpenMP** directives [22, 23]. The numerical implementation of the two phases flow physics has been benchmarked against 1D analytical solutions (e.g., Section 3.4 in [12]). The new additions to the numerical implementation are successfully benchmarked against numerical an analytical solutions [14, 15].

1.4 Phase change

We assume thermodynamic equilibrium between solid and liquid at every time step, everywhere in the model domain. Phase change is therefore not limited by reaction kinetics but only by phase diagrams (which describe the solid-liquid equilibrium) and energy conservation. Phase change is treated differently for

Fig. S3 Composition of the melt and solids in Sm/Nd (magenta), Lu/Hf (green) and Hf/W (cyan) as function of melt fraction. This plot does not involve any fluid dynamics calculation and serves as a consistency check for the behaviour of our numerical simulations (see Fig. S4). The trace elements ratio are normalized by their ratio in the bulk silicate Earth (BSE). (a and c) In the upper mantle, trace elements are so incompatible in the olivine that the composition departs from the BSE composition only towards the end of solidification. Solids however are very fractionated. (c and d) In the lower mantle, the trace elements considered have a better affinity for the bridgmanite. Liquids are progressively fractionated from the onset of solidification until the end of the crystallization.

both thermal and compositional density contrasts - we detail ~~ed~~ below where the origin of this compositional differences comes from - act in the same direction. These density contrasts drive the growth of Rayleigh-Taylor instabilities forming crystal-rich downwellings (Figure S7).

As shown in this study, crystals formed at the surface of planet play a major role in the differentiation of the mantle. Indeed, these solids experience shallow depth solid-liquid chemical fractionation governed by low-pressure mineral phases. These crystal then sink as downwellings in the deep mantle without remelting (see Section 2.3). This is the reason why the chemical evolution of the deep mantle is expected to be significantly affected by low-pressure solid-liquid partitioning. However, to bring a fractionated composition into the deep mantle, solids and liquids must segregate from each other in the vicinity of the surface (cold) thermal boundary. Otherwise, it is the bulk composition of the liquid ~~that can also be fractionated at other locations~~ - that is brought to the deep mantle.

Here, we perform a scaling analysis to compare the growth timescale of a Rayleigh-Taylor instability with that of solid-liquid segregation. First, we derive a dimensional equation to quickly estimate the timescale ratio using conservative values. Then, we reformulate it with dimensionless numbers to better relate our fluid dynamics simulation to realistic conditions.

Fig. S4 Benchmark of trace elements partitioning in the fluid dynamics simulations. Analytical predictions of both melt and solid composition (green and blue lines) estimated by a simple partitioning model (see Eq.(24) and Fig. S3). (Top) Partitioning coefficients correspond to the one of olivine. (bottom) Partitioning coefficients correspond to the one of bridgmanite. Assuming a melt fraction of 10 %, we can predict the composition of the melt and solid by determining the intersection of the dash (top) or solid (bottom) lines with a horizontal line drawn at $y=0.1$. This is in good agreement with the frequency distribution of the simulations. However, the fluid dynamics simulations provide a more complex picture of the compositional fields, revealing a distribution of composition rather than a unique composition.

The growth timescale for a Rayleigh-Taylor instability driven by a density anomaly of size R_0 in a domain of viscosity η scales as:

$$\tau_{RT} \sim \frac{\eta}{\Delta\rho_{RT}gR_0}. \quad (25)$$

where η is a function of viscosity, $\eta = \eta_s \exp(-30\phi)$ [28, 29]. Assuming that the compaction length is smaller than the thickness of the thermal boundary layer - we will come back to this assumption later - the timescale for solid-liquid segregation in this layer that we assume to be mostly solid, is controlled

Fig. S8 Same as Figure (S7) but for the lower mantle in the vicinity of the basal magma ocean (BMO). In several regions the liquid velocity (red arrows) is deflected downwards relative to the solid velocity (blue arrows), which is attributed to the downwards flux of melt driven by the density contrast between the solid and liquid in the lower mantle. The top panels show closed up views the of the fluid parcels highlighted in magenta in the bottom row.

Importantly, the solid-liquid segregation mechanism near the top boundary layer leads to solids in downwelling currents that are enriched in FeO, as shown in Figure S7. This is at odds with what one would expect from the values of partitioning coefficients. FeO solids should be consistently depleted in FeO compared to liquids. However, any crystal forming near the cold thermal boundary tends to settle, propelling FeO-rich melt upwards by mass conservation. This FeO-rich melt approaches the cold thermal boundary, cools down by diffusion, and eventually solidifies. As a result, the cold downwellings are enriched in FeO, making them thermally and chemically negatively buoyant, as seen in Figure S9.

The symmetric version of the processes described above also occurs in the deep mantle during the formation of upwelling currents at the (hot) bottom thermal boundary (Figure S10). The main difference is that in the deep mantle, melts segregate downwards due to a density crossover between the melt and solid at approximately mid-mantle depth. Near and within the hot thermal boundary layer, solid materials heat up by diffusion and may remelt. Any remaining crystal tend to float, driving FeO-rich melt downwards. Consequently, this denser FeO-rich melt accumulates at the core-mantle boundary (CMB), forming the basal magma ocean (Figure S10).